

**Title: Comparing three approaches of spatial disaggregation of legacy soil maps based on**
**DSMART algorithm**
**Authors:**
Yosra Ellili[1], Brendan Philip Malone[2], Didier Michot[3], Budiman Minasny[4], Sébastien Vincent[1],
Christian Walter[3] and Blandine Lemercier[3]
[1]UMR SAS, INRA, AGROCAMPUS OUEST 35000 Rennes, France
[2]CSIRO, Agriculture and Food, Canberra, ACT, Australia
[3]UMR SAS, AGROCAMPUS OUEST, INRA 35000 Rennes, France
[4]Sydney Institute of Agriculture, School of Life and Environmental Sciences, The University of
Sydney, NSW, Australia
**Corresponding Author:** Yosra Ellili
**Corresponding Author's Institution**: UMR SAS, INRA, AGROCAMPUS OUEST 35000
Rennes, France
**Corresponding Author's contact** (email) yousraellili91@gmail.com





**Abstract:**
Enhancing the spatial resolution of pedological information is a great challenge in the field of Digital Soil
Mapping (DSM). Several techniques have emerged to disaggregate conventional soil maps initially
available at coarser spatial resolution than required for solving environmental and agricultural issues. At the
regional level, polygon maps represent soil cover as a tessellation of polygons defining Soil Map Units
(SMU), where each SMU can include one or several Soil Type Units (STU) with given proportions derived
from expert knowledge. Such polygon maps can be disaggregated at finer spatial resolution by machine
learning algorithms using the Disaggregation and Harmonisation of Soil Map Units Through Resampled
Classification Trees (DSMART) algorithm. This study aimed to compare three approaches of spatial
disaggregation of legacy soil maps based on DSMART decision trees to test the hypothesis that the
disaggregation of soil landscape distribution rules may improve the accuracy of the resulting soil maps.
Overall, two modified DSMART algorithm (DSMART with extra soil profiles, DSMART with soil
landscape relationships) and the original DSMART algorithm were tested. The quality of disaggregated soil
maps at 50 m resolution was assessed over a large study area (6,775 km²) using an external validation based
on independent 135 soil profiles selected by probability sampling, 755 legacy soil profiles and existing
detailed 1:25,000 soil maps. Pairwise comparisons were also performed, using Shannon entropy measure,
to spatially locate differences between disaggregated maps. The main results show that adding soil landscape
relationships in the disaggregation process enhances the performance of prediction of soil type distribution.
Considering the three most probable STU and using 135 independent soil profiles, the overall accuracy
measures are: 19.8 % for DSMART with expert rules against 18.1 % for the original DSMART and 16.9 %
for DSMART with extra soil profiles. These measures were almost twofold higher when validated using
3x3 windows. They achieved 28.5% for DSMART with soil landscape relationships, 25.3% and 21% for
original DSMART and DSMART with extra soil observations, respectively. In general, adding soil
landscape relationships as well as extra soil observations constraints the model to predict a specific STU
that can occur in specific environmental conditions. Thus, including global soil landscape expert rules in
the DSMART algorithm is crucial to obtain consistent soil maps with clear internal disaggregation of SMU
across the landscape.
**Key words:** digital soil mapping, soil landscape relationships, spatial disaggregation, DSMART








1)  Introduction
Characterizing soil variability especially over large areas, remains a crucial challenge to foster
sustainable management of agronomic and environmental issues and help stakeholders to design
regional projects (Chaney et al., 2016).  At the regional as well as country level, soil maps are often
available at coarse spatial resolution (Bui and Moran, 2001) which limits their ability to depict
accurate soil information. For instance, the finest soils maps covering France were elaborated by
administrative region at 1:250,000 scale, via a set of polygons, called Soil Map Units (SMU) with
crisp boundaries. The delineation of SMU is based on soil survey programmes involving
pedologists' expertise. In a coarse scale map, each polygon includes one or several Soil Type Unit
(STU), which are not explicitly mapped, but their proportions and their environmental conditions,
as well as soil characteristics, are provided in a detailed database (Le Bris et al., 2013).
To improve soil variability knowledge and overcome the limitation of a coarse mapping scale,
several methods have emerged in the field of Digital Soil Mapping (DSM). These methods offer
useful tools to predict soil spatial pattern from scarce or limited soil datasets by exploiting the
availability of model based methods and an extensive array of spatialise (and more often than not
gridded) environmental variables. In recent decades, DSM techniques have been increasingly used
to downscale soil information and improve their spatial resolution. Depending on the quality of
data and the complexity of soil cover, Minasny and McBratney (2010) supply a workflow that
outlines different models that can be explored. In general, two main pathways can be distinguished:
point based DSM approaches and map disaggregation approaches (Odgers et al., 2014; Holmes et
al., 2015). Point DSM approaches used legacy soil profiles, which are irregularly distributed and
collected according to specific objectives rather than to optimise a statistical criterion (Holmes et
al., 2015). The spatial distribution of soil properties can be estimated by fitting geostatistical
models such as ordinary kriging (Odgers et al., 2014; Holmes et al., 2015; Chaney et al., 2016;
Vincent et al., 2018; Chen et al., 2018) or cokriging, which takes into account the spatial
interrelations among several soil properties (Webster and Oliver, 2007).  Additionally, McBratney
et al. (2003) developed the SCORPAN soil landscape model. It is an empirical quantitative function
of environmental covariates, allowing predicting soil attributes (soil type or soil property) based
on correlative and statistical relationships with predictor variables.



The second approach, known as spatial disaggregation, attempts to downscale the soil map unit
information to delineate unmapped STUs (Bui and Moran, 2001; Odgers et al., 2014; Holmes et
al., 2015). Alternatively, it can be defined as the process that allows estimating soil properties at a
finer scale than the initial soil map. Several techniques have been demonstrated through soil science
literature and tested in different case studies around the world. For instance, Kempen et al. (2009)
have explored the use of multinomial logistic regression (MLR) for digital soil mapping. Other
techniques have also been applied as decision trees using rule based induction (Bui and Moran,
2001), Bayesian techniques (Bui et al., 1999) and an area to point kriging method (Kerry et al.,
102  2012).

In the DSM field, machine learning techniques are increasingly used to elucidate the spatial
distribution of both soil type and soil properties across a large range of scale (Bui and Moran.,
2001; Scull et al., 2005; Lacoste et al., 2011; Lemercier et al., 2012; Nauman and Thompson, 2014;
Holmes et al., 2015; Vaysse and Lagacherie, 2015; Ellili et al., 2019). They were also applied to
disaggregate superficial geology maps available at 1: 250 000 scale in Australia (Bui and Moran,
2001). The main advantage of these approaches is they allow handling both quantitative and
categorical (ordinal or nominal) soil and environmental variables, as explanatory covariates (Bui
and Moran, 2001).
Odgers et al. (2014) have developed a machine learning algorithm entitled Disaggregation and
Harmonisation of Soil Map Units Through Resampled Classification Trees (DSMART) to predict
STU as a function of the high resolution environmental data supplied over different study areas in
Australia. The DSMART algorithm is based on a calibration dataset derived from a random
selection of a fixed number of sampling points within each soil polygon. Each sampling point is
then assigned to one soil type following a weighted random allocation procedure based on the
proportions informed in the soil map database. The same procedure was applied by Chaney et al.
(2016) to spatially disaggregate the soil map of the contiguous United States at a 30m spatial
resolution using petascale High Performance Computer (HPC). Because integration of pedological
knowledge has been recognized as an effective way to improve digital soil mapping approaches
(Cook et al., 1996; Walter et al., 2006; Stoorvogel et al., 2017; Machado et al., 2018; Møller et al.,
2019), Vincent et al. (2018) have applied the DSMART algorithm with additional expert soil
landscape rules describing soil distribution in the local context of the Brittany region (France). By





adding supplement sampling points to the calibration dataset selected according to soil parent
material, soil redoximorphic conditions and topographic features, and by integrating soil landscape
relationships in the DSMART sample allocation scheme, the authors obtained a coherent soil
spatial distribution observing soil organisation along hillslopes and occurrence of intensely
waterlogged soils in the stream neighbourhood, as observed in Brittany.
This study aimed to test the hypothesis that adding soil landscape relationships in the disaggregation
procedure improved the accuracy of produced disaggregated soil maps. This involves assessing the
contribution of soil landscape relationships implemented in the DSMART algorithm by Vincent et
al. (2018). To achieve this objective, we compared disaggregated soil maps either derived from the
original DSMART algorithm, the DSMART algorithm with extra soil observations and the
DSMART algorithm fed by soil landscape relationships over an area of 6,775 km² in the eastern part
of Brittany, France.

### 2) Materials and methods

2.1) Study area

The Ille et Vilaine department covers an area of 6,775 km² and is located at the eastern part of
Brittany, France (48°N, 2° W) (Fig 1). It is drained by the rivers Ille and Vilaine and their
tributaries. Its climate is oceanic, with a mean annual rainfall of 669 mm and mean annual
temperature of 11.3° (Source: Climate Data EU). Main land uses comprise arable land, temporary
and permanent grasslands, woodland, and urban areas. In the present study, anthropogenic areas
were not considered. Elevation ranges between 0_20 m in the coastal zone and 20_150 m almost
everywhere expect in the western part of the department where it tills 256 m. The topography is
generally gentle with maximum slopes not exceeding 16%. The Ille et Vilaine department is part
of the Armorican Massif with complex geology (BRGM, 2009): intrusive rocks (granite, gneiss
and micaschist) in northern and north western zones, sedimentary rocks (sandstone) and
metamorphic rocks (Brioverian schist) in the central and southern zones, and superficial deposits
(Aeolian loam with decreasing thickness from north to south overlaying bedrock, alluvial and
colluvium deposits). According to the World Reference Base of Soil Resources, soils occurring in
Ille et Vilaine include Cambisols, Luvisols Stagnic Fluvisols, Histosols, Podzols, and Leptosols
(IUSS Working Group WRB, 2014).
2.2) Soil data



2.2.1) Regional soil database at 1:250 000 scale
In Brittany, soils are represented through a regional geographic database called "Référentiel
Régional Pédologique (RRP)" available at 1:250,000 scale (INRA Infosol, 2014).This regional
database identifies soils within Soil Map Units (SMUs), each containing one to several soil types
called Soil Type Units (STUs). STUs are defined as areas with homogeneous soil forming factors,
such as morphology, geology, and climate. In the study area, 96 SMU and 171 STU have been
distinguished and represented by a spatial coverage of 479 polygons.
In the regional database, SMUs were spatially delimited with crisp boundaries, while STUs were
not explicitly mapped, but their proportion in each SMU as well as associated environmental and
soil characteristics were accurately described in a semantic database (Le Bris et al., 2013; INRA
Infosol, 2014).
2.2.2) Soil validation data
To assess the quality of disaggregated soil maps, three validation datasets were used (Fig. 1):
• 135 soil profiles chosen following a stratified random sampling design and specifically

described and sampled from March to May 2017 for independent validation purposes in the

framework of the Soilserv research project (Ellili et al., 2019, submitted).

• 755 legacy soil profiles collected between 2005 and 2008 during the "Sols de Bretagne"

programme (INRA Infosol, 2014).These profiles were sampled to characterize

hydromorphic soil conditions and soil landscape heterogeneity.

Existing detailed soil maps (1:25,000) covering 87,150 ha, surveyed according to Rivière et al.
(1992) and revised later to adapt to the STU typologies developed in the RRP (Le Bris et al., 2013).
All soil profiles were allocated after description and analysis by an expert to a suitable STU. Both
legacy soil profiles and detailed maps were converted to raster format to perfectly meet the
prediction raster at 50m spatial resolution.
2.3) Environmental covariates
The SCORPAN concept (McBratney et al., 2013) allows one to predict STU as a function of a set
of covariates describing seven soil forming factors, namely soil properties (s), climate (c),
organisms (o), relief (r), parent material (p), age (a) and geographic position (n). In this study, ten





environmental variables (Table 1) were considered as covariates in the disaggregation process at a
50m spatial resolution. Terrain attributes included elevation, slope, Compound Topographic Index
(CTI) (Beven and Kirkby, 1979, Merot et al., 1995) and Topographic Position Index (TPI) (Vincent
et al., 2018) that together were derived from a 50m resolution Digital Elevation Model (IGN, 2008).
These attributes were computed using ArcGIS 10.1 (ESRI, 2002) and MNT surf software
(Squividant, 1994).
Environmental attributes describing soil parent material (Lacoste et al., 2011) and hydromorphic
soil conditions via waterlogging index (Lemercier et al., 2013) were obtained using decision tree
methods. Waterlogging index derives from a natural soil drainage prediction. Four classes were
distinguished: well drained, moderately drained, poorly drained and very poorly drained. Aeolian
silt deposits and Soil Map Units boundaries are environmental covariates also obtained via expert
knowledge from soil scientists.
Landscape units reflecting vegetation, land use, and relief attributes were derived from a MODIS
imagery by supervised classification (Le Du Bayo et al., 2008). The Airborne gamma ray
spectrometry variable (K:Th ratio) (Messner, 2008), characterizing the degree of weathering of the
geological material, was also taken into account.
All soil environmental covariates were converted to raster format at 50 m spatial resolution.
2.4) Disaggregation procedure: DSMART algorithm

2.4.1) Original DSMART algorithm (Method 1)

The open source DSMART algorithm (Odgers et al., 2014) was applied to spatially disaggregate
the existing legacy soil map at 1:250,000 scale. DSMART algorithm uses machine learning
classification trees implemented in C5.0 (Quinlan, 1993) to build a decision tree from a target
variable (STU) and the environmental covariates supplied. The DSMART algorithm was written
in the Python programming language by Odgers et al. (2014) and was recently translated in the R
programming language.
Running DSMART algorithm requires four main steps (Fig. 2):



1) Polygon sampling by a random selection of a fixed number of sampling points (n=30)
within each polygon. This procedure allowed to select a total of 14,370 sampling points,
per iteration, covering the study area and ensured that all polygons were sampled.
2) Soil Type Unit (STU) assignment to each sampling point following a weighted random
allocation method. This step was based on the proportion of each STU informed in the RRP
database.
3) Decision tree generation: the full set of sampling points were spatially intersected with the
selected environmental covariates. This georeferenced dataset was then used as a
calibration dataset to build the decision tree allowing the prediction of an STU as a function
of environmental covariates. C5.0 created explicit models, which were applied to the
covariates rasters to generate a realisation of STU distribution over the study area at 50 m
resolution.
These three steps were repeated 100 times to generate 100 realisations of the potential soil type
distribution over the study area at 50 m of resolution.
4) Computing the probabilities of occurrence: the 100 realisations were stacked to calculate
the probability of occurrence of each predicted STU by counting the frequency of each STU
at each pixel. This procedure led to a set of 171 rasters depicting the probability of
occurrence of 171 STU.

2.4.2) Original DSMART algorithm + soil observations (method 2)

This disaggregation approach is similar to the original DSMART algorithm. However, the main
difference is that 755 additional soil profiles, spatially collocated, were added to the calibration
dataset to build decision trees. These soil profiles make it possible to incorporate real field
observations with established soil landscape relationships. For each realisation, a calibration
dataset (15, 125 samples) including virtual samples randomly selected from polygon units, as well
as soil observations were used to model soil type with environmental covariates. The model was
then extrapolated over the study area.

2.4.3) Original DSMART algorithm + expert rules (Method 3)



Including soil landscape relationships in the disaggregation process was explored by Vincent et al.
(2018) in a specific regional pedoclimatic context in Brittany (France). Expert soil landscape
relationships were used to assign STU to sampling points. These relationships were based on expert
pedological knowledge, which takes into account soil parental material as well as topography and
waterlogging in the UTS allocation procedure. This approach combines two sources of the dataset
to calibrate the model. The first one was derived from semantic information for each SMU/STU
combination. It consists in attributing a barcode to each SMU/STU combination, derived from a
concatenation of four features contained in the RRP database (parent material, SMU identifier, TPI
and waterlogging index), and to compare these barcodes to a stack of regional covariates
representing the same four features, to assign each pixel of the study area to a suitable STU. This
procedure allowed matching soil exhibiting specific features with their potential spatial
distribution. For instance, hydromophic soils occur with slope sequences and valley positions,
while well drained soils occur in upslope or middle slope positions. Using a random sampling
stratified by SMU's area, a set of sampling points was selected with a proportion of one sample for
every 5 hectares and a minimum of five samples per polygon unit.
The second dataset was derived from a random sampling of a fixed number of sampling points in
each polygon unit. This procedure ensured that all polygons had been sampled. STU allocation was
based on the soil map unit proportions. The full set of each realisation (18, 320 samples) combining
expert calibration dataset as well as dataset derived from random sampling procedure was spatially
intersected with existing environmental covariates and used as a unique calibration dataset to build
decision trees.

2.4.4) Prediction of the most probable STUs
From all soil type probability rasters obtained, only the three most probable STUs (with the highest
probability of occurrence) were considered: for each pixel, the final prediction was the combination
of the three most probable predicted STUs (1st STU, 2sd STU, and 3rd STU) and their associated
probability of occurrence.
The classification confusion index (CI) between the first most probable STU and the second most
probable STU was calculated following Eq.1:
$CI = 1 - (P_{1^{st} STU} - P_{2^{sd} STU})$ [1]





Where $P1^{st}_{STU}$ and $P2^{sd}_{STU}$ denote respectively the highest probability of occurrence for $1^{st}$ STU
and the second highest probability of occurrence for $2^{sd}$ STU, calculated at each pixel (Burrough et
al., 1997; Odgers et al., 2014).
This index was considered as an indicator of certainty assessment about the most probable
predicted soil class and is ranging between 0 and 1. It tends to 1. When the $1^{st}$ STU and $2^{sd}$ STU
are predicted with similar probability of occurrence and zero when the probability of occurrence
of the $2^{sd}$ STU is close to zero.

2.5) Validation of disaggregated soil maps
The quality of soil maps resulting from the three DSMART algorithm based approaches was
assessed by combining both spatial and semantical validation methods. Spatial validation is divided
into 2 sub approaches ("pixel to pixel" and "window of 3x3 pixels"). For detailed soil maps and
accurate soil profiles, "pixel to pixel" validation consists in checking, at each pixel, if the predicted
STU respects the observed STU value (Heung et al., 2014; Nauman et al., 2014; Chaney et al.,
2016; Møller et al., 2019). The "window of 3x3 pixels" validation assumes that, for each pixel, the
predicted STU respects the observed STU value if it matches at least one of its 9 surrounding
neighbours (Heung et al., 2014; Chaney et al., 2016). This method provides some flexibility by
compensating spatial referencing error of soil maps and avoids the impact of fine scale spatial
noise.
The semantical validation was also performed considering either each STU or a group of STUs
sorted by expert on the basis of similar pedogenesis factors and similar diagnostic horizons
(Vincent et al., 2018; Møller et al., 2019). From the initial 171 STUs described in the soil database,
the sorting procedure led to 78 groups and 11 STU remained single.

2.6) Pair wise comparisons of disaggregated soil maps
To compare the soil type rasters derived from the three DSMART based approaches, pairwise
comparisons were performed using *Vmeasure* method implemented as open source software in an
R package called Spatial Association Between REgionalisations (SABRE) (Rosenberg and
Hirschberg, 2007). This is a spatial method developed to compare maps in the form of vector
objects and it was commonly used in computer science to compare (non spatial) clustering.



We divide the entire study area into 2 different sets of regions, referred to as regionalizations R and
Z. The first regionalization R divides the domain into n regions $r_i$ (i=1 to n) and the second
regionalization Z divides the domain into m zones $z_j$ (j=1 to m). Superposition of the 2
regionalization R and Z divides the domain into n x m segments having $a_{ij}$ area. The total area of a
region $r_i$ is $A_i = \sum_{j,1}^{m} a_{ij}$, the total area of a zone $z_j$ is $Aj = \sum_{i,1}^{n} a_{ij}$ and the total of the domain is
$A = \sum_{j=1}^{m} \sum_{i=1}^{n} a_{ij}$.
The SABRE package calculates a degree of spatial agreement between two regionalizations using
an information theoretical measure called the *V measure*. *V measure* provides two intermediate
metrics: *homogeneity* and *completeness*. *Homogeneity* is a measure of how well regions from the
first map fit inside zones from the second map (Eq 2). *Completeness* measures how well zones
from the second map fit inside regions from the first map (Eq 5). The final value of *V measure* is
calculated as the weighted harmonic mean of homogeneity and completeness (Eq 8). All metrics
range between 0 and 1, where larger values indicate better spatial agreement. *V measure*,
homogeneity, and completeness are global measures of association between the two
regionalizations.
Additional indicators of disaggregation quality were calculated using Shannon entropy index of
regions and zones (Shannon 1948; Nowosad and Stepinskie, 2018). These indicators qualify local
associations by highlighting the region's inhomogeneities (Eq 3, Eq 4), or zone's inhomogeneities
(Eq 6, Eq 7). Two normalized Shannon entropy was also computed using the ratios $(S_j^R/S^R)$ and
$(S_i^Z/S^Z)$ to derive maps of local spatial agreement between the two regionalizations R and Z. These
measures have a range between 0 and 1.
When $S_j^R$ (Eq3) is close to zero, this denotes that the zone j is homogenous in terms of regions
(each zone is within a single region). However, when $S_j^R$ value increases the zone is increasingly
inhomogeneous in terms of regions (it overlays an increasing number of regions). Therefore, $S_j^R$
(Eq 3) assesses the degree of this inhomogeneity or a variance of region in zone *j*. A global indicator
that measures a homogeneity of a given zone in terms of regions is given via Eq 2.
Analogous to homogeneity but with the roles of regions and zones reversed, the dispersion of zones
over the entire area is also computed using Shannon entropy (Eq 4 and Eq 7), and a global indicator
*C* (Eq 5) measures a homogeneity of a given region in terms of zones.





$$h = 1 - \sum_{j=1}^{m} \left(\frac{A_j}{A}\right) \left(\frac{Variance\ of\ regions\ in\ zone_j = S_j^R}{Variance\ of\ regions\ in\ the\ domain = S^R}\right)$$ [2]
$$S_j^R = -\sum_{i=1}^{n} \left(\frac{a_{i,j}}{A_j}\right) \log \left(\frac{a_{i,j}}{A_j}\right)$$ [3]
$$S^R = -\sum_{i=1}^{n} \left(\frac{A_i}{A}\right) \log \left(\frac{A_i}{A}\right)$$ [4]
$$c = 1 - \sum_{i=1}^{n} \left(\frac{A_i}{A}\right) \left(\frac{Variance\ of\ zones\ in\ region_i = S_i^Z}{Variance\ of\ zones\ in\ the\ domain = S^Z}\right)$$ [5]
$$S_i^Z = -\sum_{j=1}^{m} \left(\frac{a_{i,j}}{A_i}\right) \log \left(\frac{a_{i,j}}{A_i}\right)$$ [6]
$$S^Z = -\sum_{j=1}^{m} \left(\frac{A_j}{A}\right) \log \left(\frac{A_j}{A}\right)$$ [7]
$$V_\beta = \frac{(1+\beta)hc}{(\beta h)+c}$$ [8]
β is a coefficient that allows promoting the first or the second regionalization, and by default, β
equals 1. $V_\beta$ has a range between 0 and 1. It equals 0 in case of no spatial association and 1 in case
of perfect association.
The *V measure* method was applied in two main situations (DSMART+expert rules, Original
DSMART) and (DSMART + expert rules, DSMART+extra soil observations). The reference map
is always the map derived from DSMART algorithm with expert soil landscape relationships.
3)  Results

3.1) Disaggregated soil maps

Applying DSMART based approaches yielded a set of soil maps and associated probability of
occurrence rasters. The original DSMART approach allowed to disaggregate the 96 SMUs into
108 STUs while DSMART with expert rules approach yielded 158 STUs and DSMART with extra
soil observations approach yielded 172 STUs with respect to the first most probable STU map. A
total of 171 STUs were identified in the Ille et Vilaine department within the RRP database.
Unpredicted STUs correspond mainly to rare STUs with low proportions ranging between 2 and
10% within the SMUs containing them.
Figure 3 shows the three maps of the 1[st] most probable STU derived from each approach as well
as the original soil map. Overall, the three most probable STUs maps captured the main pattern of



soil distribution of the coarse soil map. As one could expect according to the geological parent
material map (Lacoste et al., 2011), extensive areas of deep silty soils are developed in Aeolian
loam deposits encountered in the north east as well as in the north central parts of the study area.
Colluvial and alluvial soils were mainly predicted in the north coast part and large valleys zones.
The visual comparison of disaggregated soil maps highlighted global similarities in the soil spatial
distribution markedly affected by SMU boundaries. The three approaches distinguished very well
soils developed in marsh parent material in the coastal part (north) of the study area. However,
DSMART with soil landscape expert rules map as well as DSMART with extra soil observations
map remained more detailed and underlined a clear internal disaggregation of SMUs especially in
the north and the central parts of the Ille et Vilaine department. Visual inspection of the obtained
DSMART with extra soil observations map as well as DSMART with expert rules map showed an
increase in soil heterogeneity when compared to Original DSMART map. More importantly,
legacy soil profiles made it possible to take into account some rare soil types with low probability
to be predicted. Therefore, adding supplement sampling points via the expert calibration dataset
and the 755 extra soil profiles allowed to predict STUs characterized in the soil database with a
low spatial extent. Nevertheless, the three DSMART based approaches spatially disaggregated the
most frequent components disregarding the less frequent ones.
Figure 4 shows maps of the global probability of redoximorphic soils across the study area. STU
probability rasters, depicting hydromorphic soils, were added together to produce continuous maps
of hydromorphic soil probability. Visual inspection of three maps highlighted global similarities,
but local differences were recorded along the hydrographic network and in the southern part of the
study area. As could be expected, DSMART with expert rules well predicted hydromorphic soils
in valleys and coastal areas, with a probability of occurrence exceeding 80%. Adding soil landscape
relationships in the allocation process constrained hydromorphic soil predictions in specific
landscape positions. The same trend characterized DSMART with extra soil observations map,
particularly in the central part of the study area. Therefore, including 755 soil profiles had an
important role in the disaggregation process in the northern and the central parts where these
profiles were located.
The quality of maps resulting from DSMART based approaches was quantified via the probabilities
of occurrence of each STU predicted and the confusion index maps (Fig. 5). The latter measure



indicated areas where the probability of occurrence of the two most probable soil types was close.
Over the study area, the average probability of occurrence of the most probable soil type achieved
respectively 0.41 for DMSART map, 0.68 for DMSART with expert rules and 0.28 for DSMART
with extra soil observations maps. Meanwhile, the average confusion index reached 0.8 for the
original DSMART approach while DSMART with extra soil observations and DSMART with
expert rules achieved 0.9 and 0.43, respectively. Although the most probable soil classes provide
plausible maps of soil distribution, there is a significant prediction uncertainty as depicted by these
measures.
In regions where disaggregated soil maps showed low confusion index, particularly in northwest
and north coast areas of Ille et Vilaine department, high confidence in predictions was recorded.
These areas were predominantly deep loamy soils or developed in alluvial and colluvium deposits.
Figure 6 compares the cumulative area of the STUs estimated from the three disaggregated maps
and that derived from the regional soil database. For each STU, its relative predicted area was
estimated by counting the number of pixels where it was predicted. For the regional soil database,
each STU area was computed from total SMU area multiplied by the proportion of the STU. This
comparison shows that some STUs were overestimated by the disaggregation approaches when
comparing to the soil database. DSMART with extra soil observations and original approaches
showed similar cumulative STU areas under the curve whereas DSMART with expert rules had a
shape similar to the regional soil database.
The most abundant STU in the database (431: Stagnic Fluvisol developed from alluvial and
colluvium deposits) was predicted as the most frequent STU by DSMART with extra soil
observations and DSMART with expert rules, and it was predicted as the second most abundant
STU by the original DSMART algorithm. The 10 most abundant STUs in the soil database covers
almost 43% of the study area. Of them, 7 belong to the 10 STU most predicted by the three
disaggregation approaches (Table 2).
3.2) Covariates importance in the decision trees
Figure 7 gives the relative importance of the covariates used in DSMART based approaches. Soil
parent material and SMU boundaries were used systematically in condition rules regardless of the
disaggregation method. This was consistent with the contrasting pattern of geology and the





dependence relationship between SMU and its soil components. Considering the original
DSMART approach (Fig. 7.a), distribution functions of Aeolian silt deposits, airborne gamma ray
spectrometry variable (K:Th ratio) and elevation contributions were more dispersed according to
the STU considered than those of other covariates. For instance, Aeolian silt deposits contribution
varied between 20 and 80% with a median value of 42%, whereas slope contribution ranged
between 20 and 40 % with a median value of 28%. Aeolian silt deposits have an important weight
in STU predictions, due to its ability to represent soils inherited from this superficial parent
material, which is poorly represented in lithological maps.
DSMART with soil landscape relationships (Fig. 7.b) showed almost the same distribution function
of all covariates except for elevation where its distribution function was more dispersed. Since a
part of training samples was chosen with expert knowledge based on three environmental
covariates: TPI, a waterlogging index and soil parent material, we would expect the prominent role
of waterlogging index and TPI to constrain hydromorphic soils predictions and to achieve STU
distribution in the appropriate order along the toposequence. This most likely explains the
dominance of Fluvisol Stagnic in valleys areas followed by a transition to Cambisols commonly
found at upslope and midslope positions along the toposequences.
Analogous to the original DSMART algorithm, DSMART with extra soil observations (Fig. 7.c)
highlighted almost the same distribution of use of soil environmental covariates in the decision
trees, except for aeolian silt deposits, K:Th ratio and elevation. The latter covariates contributions
remained less dispersed compared to the original DSMART approach.
3.3) Validation of disaggregated soil maps
The validation procedure was performed for each DSMART based approach applied, considering
the three most probable soil types and using both semantic objects (STU or soil group) and spatial
neighbourhood (per pixel or 3x3 window of pixels).
Considering 755 legacy soil profiles prospected in the framework of "Sols de Bretagne" project,
per pixel validation accuracy reached 27%, for original DSMART maps and 34 % for DSMART
with expert rules (Table 3). A similar comparison using 135 validation sites derived from Soilserv
project showed that 18.1 % of soil profiles match DSMART maps, 19.8 % match DSMART with
expert rules maps and only 16.9 % match DSMART with extra soil observations maps (Table 3).



Using a 3 x 3 window of pixels markedly improves the global accuracies, which increased for the
two validation datasets (Table 3). DSMART with soil landscape relationships remained the best
performing method.
When compared to accurate soil maps (1:25,000), the validation procedure showed that DSMART
with extra soil observations as well as DSMART with soil landscape expert rules had almost the
same performance (37% and 38%) while best accuracy (44%) was observed for Original DSMART
maps (44%) (Table 3). These scores were clearly improved by considering soil groups and 3x3
pixels neighbourhood. For instance, the accuracy of DSMART with expert rules maps using soil
group reached 45.9% and increased to 62.1 % when considering 3x3 pixels windows (Table 3).
3.4) Comparing disaggregated maps
Figure 8 shows inhomogeneity maps measured by Shannon entropy. The map derived from
DSMART with soil landscape relationships was chosen as a reference map. This map deeply
disaggregates the initial SMUs into 120,653 regions with irregular shapes. By contrast, Original
DSMART map remained very similar to the original map and delineated the study into 40,459
regions. Both disaggregated maps reflect the main pattern of soil distribution over the study area
despite the difference in the disaggregation process. Visual inspection of maps DSMART with soil
landscape rules map and Original DSMART map revealed an overall similarity between
disaggregated maps, but local differences between them were depicted.
We calculated $h_1 = 0.49$, $c_1 = 0.58$ and $V_1 = 0.53$ as global measures of spatial agreement between
the two maps (DSMART+expert rules and Original DSMART). The average homogeneity of the
DSMART with soil landscape rules map with respect to the Original DSMART map was qualified
via $h$ homogeneity index. Similarly, the average homogeneity of the Original DSMART map with
respect to the DSMART with soil landscape rules map was qualified via $c$ completeness index.
Visually, the Fig. 8.b map seemed to be more homogeneous than the map Fig. 8.a in agreement
with the statistical assessment $c > h$. The large number of DSMART with soil landscape rules map
regions, which was three times higher than Original DSMART map zones, might explain this
difference. It is more likely that DSMART with soil landscape rules map regions cross through
multiple Original DSMART map zones than vice versa. However, two disaggregated maps
remained spatially associated according to the high $V_1$ score. The two inhomogeneity maps (Figs.

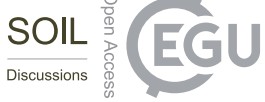

8a and 8b) highlighted the locations of greatest differences between two maps, mainly along the
hydrographic network.

When comparing disaggregated soil maps derived from modified DSMART algorithm (DSMART
with soil landscape rules and DSMART with supplement soil observations), we note that the
DSMART with extra soil observations map delineated the study area into 132,942 regions. For
both maps, internal disaggregation was well pronounced expect for DSMART with extra soil
observations map in the southern part of the study area. Visual inspection of selected maps showed
high spatial agreement and highlighted some locations of greatest differences, particularly in the
southern part of the Ille et Vilaine department. Even if the hydrographic network was well detailed
in both maps, it appeared more developed in DSMART with extra soil observations soil map.
Applying *V measure* method for assessing the spatial similarity between DSMART with soil
landscape rules map and DSMART with supplement soil observations map provided similar
information theoretical measures $h_2 = 0.47$, $c_2 = 0.48$, and $V_2 = 0.47$. Visual comparison of soil
inhomogeneity maps revealed constant variance measured by normalized Shannon entropy. This
was in agreement with the quantitative assessment $c = h$. Overall, the two disaggregated maps were
spatially correlated, as indicated by the global spatial agreement measure $V_2$.

4) Discussion

4.1) Performance of the disaggregation procedures


Produced disaggregated soil maps closely resemble the abundant soils in the original soil map
(Holmes et al., 2015; Fig.3). The 1[st] most probable STU map derived from DSMART based
approaches captured the main spatial pattern of soil distribution across the study area. More internal
variation within SMUs was found when using DSMART with added point observations and
DSMART with soil landscape relationships. Local soil heterogeneity reflecting inherent
pedological complexity was depicted by the 1[st] STU maps which deliver a deterministic soil
landscape distribution, continuously varying with landscape features.
External validation was performed to assess the quality of disaggregated soil maps. Using 135
independent soil profiles and a per pixel validation approach, the overall accuracy reached 18.1%
for DSMART algorithm 1[st] STU map, 19.8% for DSMART with expert rules 1[st] STU map and





16.9% for DSMART with extra soil profiles 1st STU map. In the DSM literature, researchers who
applied classification tree decision methods founded similar validation results. For instance, by
applying DSMART algorithm in eastern Australia and using 285 legacy soil profiles, Odgers et al.
(2014) achieved an overall accuracy of 23%. Similarly, Nauman and Thompson (2014) explored
the use of expert rules for soil landscape relationships in the United States and achieved global
accuracy ranged between 22% and 24%. Similar disaggregation performance was recorded by
Holmes et al. (2015) in Western Australia (20%), Chaney et al. (2016) in the United States (17%)
and Møller et al. (2019) in Denmark (18%) using DSMART algorithm (Table 4). In contrast to the
latter studies, a large number of STU (171 STU) compose our soil dataset. This could certainly
decrease the chance of predicting the right STU, even through mobilizing relevant geographic
dataset to implement soil landscape relationships.

When considering a window of 3x3 pixels, the overall accuracy increased considerably for the
three DSMART based approaches maps, but DSMART with expert soil landscape relationships
achieved the highest accuracy scores. Chaney et al. (2016) highlighted a high degree of spatial
noise in the predictions by including pixel validation neighbours. Overall, prediction accuracy
increased twofold with a 3x3 pixel validation window and when grouping soils to a coarser level
of soil classification (171 versus 89 soil group). This was recorded for all disaggregated maps
regardless of the disaggregation procedure and suggests that fine soil taxonomic dissimilarities can
not be accurately mapped by disaggregation processes.

4.2) Legacy soil data

Legacy soil data used in this study provide an overall representation of soil over large areas (1:
250,000 scale). This database was derived from several soil surveys and pedological expert
knowledge. SMUs were spatially delineated, and their spatial organisation, as well as STUs
features, were described according to available soil data and pedological expertise. STUs and their
associated landscape characteristics were identified as accurately as possible using legacy soil
profiles collected according to a not probabilistic sampling design between 1968 and 2012. Hence,
differences in survey methods covering a large area over a long sampling period could lead to
errors in the STU definition or uncertainties in the estimation of their area in a given SMU.



Moreover, soil survey intensity was not uniform within SMUs. Thus, SMU components may be
derived from the unequal representation of soil samples across SMUs.
Harmonising soil data to reduce the number of STU is a great challenge by itself. Grouping some
STUs regarding their pedological similarities such as sharing comparable morphological criteria,
having similar pedogenic horizons and occurring in analogous environmental conditions is
worthwhile to be investigated. More importantly, unifying soil data according to more functional
aspects such as soil agricultural potential allows also to generate a relevant regional soil database
easily handled by soil users to satisfy their needs. Many countries around the world have already
harmonized their soil databases such as Denmark and Australia, where high pedological
complexity was captured with a reasonable STU number, with not exceeding 23 soil groups in
Denmark (Møller et al., 2019) and 73 soil groups in Australia (Holmes et al., 2015).

4.3) Taxonomic similarities

In the recent DSM literature, DSMART approach is considered as an efficient tool to disaggregate
existing coarse soil maps. In this study, we compared variants of the DSMART based approach,
which differed by the training dataset used to calibrate the C5.0 model and the allocation procedure.
Modified DSMART algorithms used additional calibration datasets derived from supplement soil
observations and expert sampling of polygons. Hence, taxonomic similarities were not taken into
account neither in the calibration process nor in the current component assignment scheme. Even
if there is a large number of STUs addressing inherent soil landscape heterogeneity, there is most
likely a short taxonomic distance between many of them. As a result, these STUs may have similar
forming conditions, making it a challenge to suitably constrain the prediction probabilities using
DSMART algorithm. This likely explains the high confusion index scores recorded in the present
study, particularly for original DSMART and DSMART with extra soil profiles approaches. As
demonstrated by Minasny and McBratney (2007), including taxonomic distance in decision trees
using pedological knowledge is a relevant way to decrease the misclassification error.  Therefore,
future effort and improvements of the DSMART algorithm should take into account the taxonomic
distance between STU in the disaggregation procedure.

4.4) Mapping comparison




A quantitative comparison between disaggregated soil maps was performed using a novel approach
called *V measure* method. This method was commonly used to assess the spatial agreement
between land cover maps and thematic biotic and abiotic factors maps, as done by Nowosad and
Stepinski (2018) in the United States, but never before for soil maps.
In the present study, $V_1$ (0.53) was larger than $V_2$ (0.47) suggesting that DSMART with expert soil
landscape relationships map is much more similar to Original DSMART map than DSMART with
extra soil observations map. This might be explained by the allocation procedure for training
samples. The original DSMART algorithm tends to promote most abundant STUs with high
proportions of occurrence within polygons and penalized STUs with low proportions (comprise
between 2 and 10%). Therefore, frequent STUs are more likely to be predicted rather than rare
STUs. Meanwhile, by adding supplement soil profiles, preliminarily assigned to a suitable STU to
the training dataset, we constrain STUs with low proportions of occurrence predictions.
Major differences between DSMART with expert rules map and DMSART with soil observations
were mainly observed in the southern part of the study area and valleys areas. In general, Fluvisol
Stagnic soils were overestimated by DSMART with extra soil observations. This was likely due to
the purposive sampling design followed to supplement soil observations. The 755 legacy soil
profiles were selected to characterize hydromorphic soil conditions and to characterize inherent
soil landscape variability supposed to be organized along the hillslope.

4.5) Improvements and future work

Even though this work emphasizes the contribution of pedological knowledge in the disaggregation
process, other pathways can also be explored to improve map's accuracy. As recommended by
Mulder et al. (2016), compensating the temporal changes and differences in laboratory analytics is
a good option to improve the quality of legacy soil data. This suggests harmonising local soil
database and regrouping some STUs with similar soil forming factors through statistical modelling.
Moreover, additional environmental covariates with high spatial resolution should be used to
capture micro landscape variability (Lacoste et al., 2014; Odgers et al., 2014; Chaney et al., 2016;
Møller et al., 2019). For example, adding a more detailed Digital Elevation Model allowed to
capture small terrain features, where may be particular, STUs occurs. Improving both polygon





sampling procedure and current components assignment scheme turned out to be important to
reduce uncertainty prediction. This suggests drawing virtual soil samples proportionally to
polygons areas and using supplement STU characteristics based on surveyor observations (slope
shape, hillslope position, soil texture …) to guide STU allocation procedure (Møller et al., 2019).
Assuming that the decision tree can be built to relate STU descriptors to legacy soil data, this
method can replace weighted random allocation procedure and should help minor STU prediction
by constraining raster probabilities.
5) Conclusion

We applied three DSMART based approaches, including original DSMART algorithm, DSMART
with extra soil observations and DSMART with soil landscape relationships, to disaggregate legacy
soil polygons over a large area in Brittany (France). Regardless of the disaggregation approach, the
produced soil maps at 50 m spatial resolution successfully address the main soil spatial pattern
regarding prior pedological knowledge of our study area. Performance assessed against 135
independent soil profiles, 755 legacy soil profiles, and accurate 1:25,000 soil maps highlighted that
DSMART with expert rules maps achieved highest validation measures. Overall, modified
DSMART algorithms allowed minor STUs prediction, whereas original DSMART algorithm
promoted abundant STUs prediction with poor spatial structure improvement. Adding pedological
knowledge as well as extra soil observations in the prediction process constrained STU
probabilities, even STUs with low proportions. However, some particular STUs reflecting
hydromorphic soils or loamy soils were greatly overestimated for all the three DSMART based
approaches.
Soil maps produced using the original DSMART and DSMART with expert rules have a high
spatial agreement, but the latter map appeared more detailed and provided a spatially continuous
and consistent STU's prediction. Therefore, generalizing soil landscape relationships taken to
account several STU descriptors and landscape features should be implemented in the future
version of DSMART algorithm to capture soil landscape heterogeneity and consequently guarantee
coherent variability of soil properties.





Acknowledgments
The authors gratefully acknowledge all farmers at the Ille et Vilaine site involved in our research.
We thank the technical staff who actively participated in field sampling and laboratory analysis.
This research was performed in the framework of the INRA "Ecoserv" metaprogram. This work
was also supported by Sols de Bretagne project and Soilserv program funded by ANR (Agence
Nationale de la Recherche) (ANR-16- CE32-0005-01).





























**Figure captions**

Figure 1: Location of the study area and the validation datasets
Figure 2: Schematic of the DSMART based approaches algorithm. The steps in DSMART are: 1)
construct the calibration dataset; 2) train C5.0 model;    3) estimate STU maps and their associated
probabilities of occurrence
Figure 3: Digital soil map of the most probable STU and their associated probability of occurrence
for the whole study area and for a focus zone, a) Legacy soil map: most probable STU for each
SMU, b) original DSMART approach; c) DSMART with expert rules; d) DSMART with extra soil
observations
Figure 4: Global probability of hydromorphic soils over the study area derived from a) original
DSMART, b) DSMART with soil landscape relationships and c) DSMART with extra soil
observations. The probabilities of the three STU with highest prediction occurrence are summed if
they are hydromorphic
Figure 5: Confusion index maps for a) Original DSMART approach; b) DSMART with expert
rules; c) DSMART with extra soil observations
Figure 6: Cumulative area of the 171 STUs estimated from the regional soil database and predicted
by different DSMART based approaches
Figure 7: Violin plots of the relative importance of each environmental covariate used in a) Original
DSMART approach; b) DSMART with expert rules; c) DSMART with extra soil observations
Figure 8: Spatial association between disaggregated maps of Ille et Vilaine department. a) map of
inhomogeneity of DSMART with soil landscape relationships map in terms of original DSMART
map b) map of inhomogeneity of original DSMART map in terms of DSMART with soil landscape
relationships map c) map of inhomogeneity of DSMART with soil landscape relationships map in
terms of DSMART with extra soil observations map d) map of inhomogeneity of DSMART with
extra soil observations map in terms of DSMART with soil landscape relationships map.
Inhomogeneity (variance) is measured by normalised Shannon entropy





**Table headings**

Table 1. Description of the environmental covariates selected. Summary of environmental covariates. P: parent material; S: soil properties; R: relief; O: Organisms; C: categorical; Q: quantitative.

Table 2. Ten most extended STUs according to the regional soil database and their respective rank by area using three DSMART based disaggregation procedures

Table 3. Overall accuracies (%) obtained using various external validation approaches for the three most probable STU

Table 4: Comparison between the size areas covered, number of soil map units, soil type units of the original legacy soil maps and the accuracy achieved in other studies using DSMART algorithm





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






●  755 soil profiles (Sols de Bretagne project)
●  135 soil profiles (Soiserv project)
▮  Accurate soil maps
▭  Study area

*Figure 1: Location of the study area and the validation datasets*



*Figure 2: Schematic of the DSMART based approaches algorithm. The steps in DSMART are: 1) construct the calibration dataset; 2) train C5.0 model; 3) estimate STU maps and their associated probabilities of occurrence*



*Figure 3: Digital soil map of the most probable STU and their associated probability of occurrence for the whole study area and for a focus zone, a) Legacy soil map: most probable STU for each SMU, b) original DSMART approach; c) DSMART with expert rules; d) DSMART with extra soil observations*





*Figure 4: Global probability of hydromorphic soils over the study area derived from a) original DSMART, b) DSMART with soil landscape relationships and c) DSMART with extra soil observations. The probabilities of the three STU with highest prediction occurrence are summed if they are hydromorphic.*





*Figure 5 Confusion index maps for a) Classic DSMART approach; b) DSMART with expert rules; c) DSMART with extra soil observations*





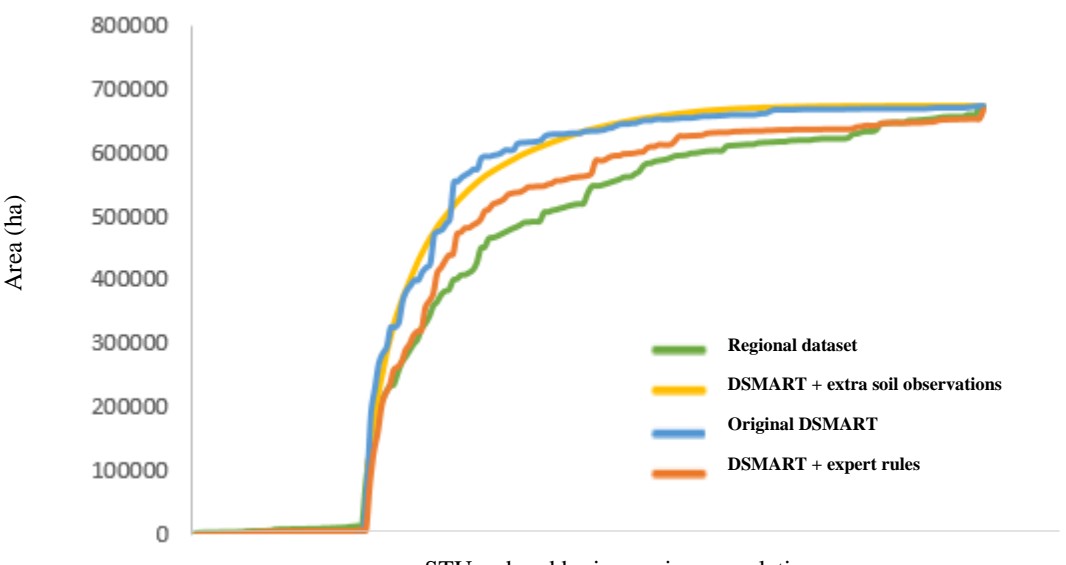

*Figure 6: Cumulative area of the 171 STUs estimated from the regional soil database and predicted by different DSMART based approaches*



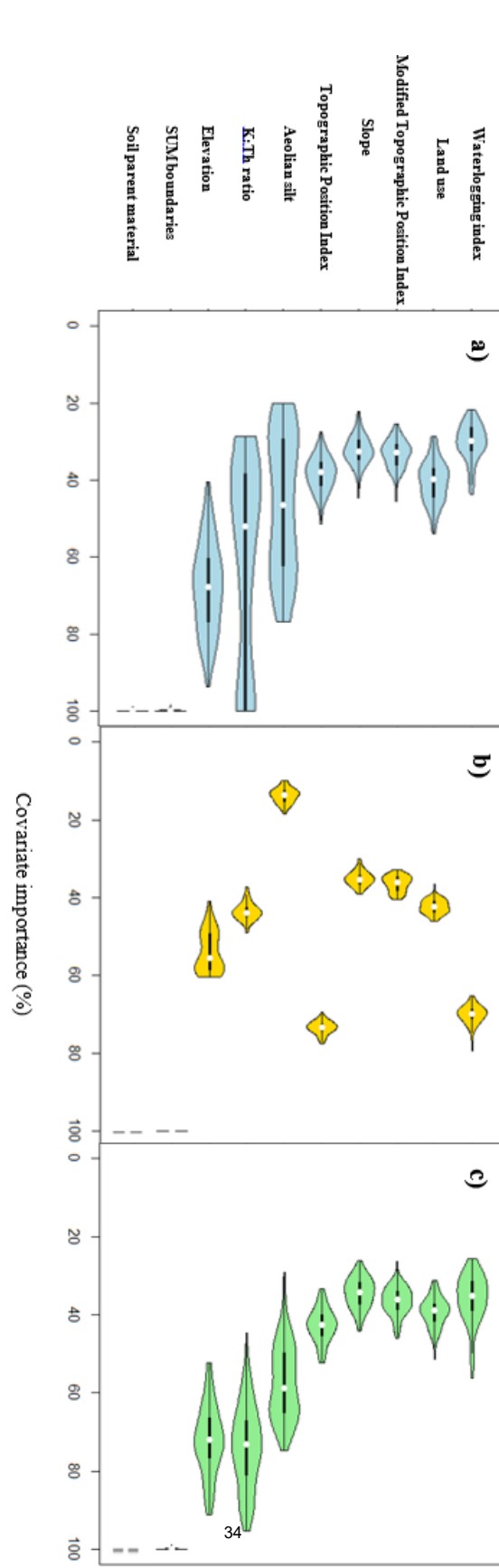

*Figure 7: Violin plots of the relative importance of each environmental covariate used in a) Original DSMART approach; b) DSMART with expert rules; c) DSMART with extra soil observations*





*Figure 8: Spatial association between disaggregated maps of Ille et Vilaine department. a) map of inhomogeneity of DSMART with soil landscape relationships map in terms of original DSMART map b) map of inhomogeneity of original DSMART map in terms of DSMART with soil landscape relationships map c) map of inhomogeneity of DSMART with soil landscape relationships map in terms of DSMART with extra soil observations map d) map of inhomogeneity of DSMART with extra soil observations map in terms of DSMART with soil landscape relationships map. Inhomogeneity (variance) is measured by normalised Shannon entropy*



*Table 2 Ten most extended STUs according to the regional soil database and their respective rank by area using three DSMART based disaggregation procedures*

| STU | | | 1:250,000 dataset | | Original DSMART approach | | DSMART with extra soil profiles | | DSMART with expert rules | |
|---|---|---|---|---|---|---|---|---|---|---|
| Label | WRB classification | Parent material | Rank | Estimated area (km²) | Rank | Predicted area (km²) | Rank | Predicted area (km²) | Rank | Predicted area (km²) |
| 431 | Fluvisol Stagnic | Alluvial and colluvial deposits | 1 | 688 | 2 | 757 | 1 | 983 | 1 | 740 |
| 248 | Cambisol | Brioverian schists | 2 | 480 | 1 | 1154 | 2 | 461 | 2 | 492 |
| 51 | Cambisol | Brioverian schists | 3 | 402 | 5 | 397 | 4 | 395 | 3 | 424 |
| 61 | Cambisol | Gritty schists | 4 | 227 | 9 | 177 | 30 | 53 | 14 | 128 |
| 183 | Cambisol Stagnic | Sandstone | 5 | 216 | 11 | 162 | 5 | 308 | 10 | 192 |
| 256 | Cambisol | Aeolian loam | 6 | 200 | 6 | 385 | 3 | 418 | 6 | 314 |
| 286 | Cambisol Stagnic | Brioverian schists | 7 | 179 | 23 | 62 | 9 | 187 | 24 | 80 |
| 86 | Cambisol | Brioverian schists | 8 | 169 | 12 | 126 | 15 | 124 | 4 | 358 |
| 340 | Albeluvisol Stagnic | Granite and gneiss | 9 | 168 | 7 | 347 | 10 | 177 | 11 | 189 |
| 54 | Cambisol | Brioverian schists | 10 | 167 | 4 | 451 | 18 | 98 | 5 | 324 |



*Table 1. Description of the environmental covariates selected*

*Summary of environmental covariates. P: parent material; S: soil properties; R: relief; O: Organisms; C: categorical; Q: quantitative.*

| Environmental covariate | SCORPAN factor | Type | Unit or number of classes |
|---|---|---|---|
| **Terrain attributes derived from the digital elevation model** | | | |
| Elevation | R | Q | m |
| Slope | R | Q | % |
| Compound Topographic Index (TPI) | R | Q | Log (m$^3$) |
| Topographic Position Index | R | C | 5 classes |
| **Pedology and geology** | | | |
| Soil parent material | P | C | 22 classes |
| Soil Map Units | R | C | 96 classes |
| Aeolian silt deposits | P | C | 2 classes |
| Waterlogging index | S | C | 4 classes |
| **Organism** | | | |
| Landscape units | O | C | 19 classes |
| **Gamma ray spectrometry from 250 m airborne geophysical survey interpolations** | | | |
| K:Th ratio | P | Q | |

*Table 3. Overall accuracies (%) obtained using various external validation approaches for the three most probable STU*

| Pixel to pixel validation of STU | | | | | |
|---|---|---|---|---|---|
| | DSMART approach | Most probable STU | Second most probable STU | Third most probable STU | Total |
| Soil maps (87 150 ha) | Original DSMART | 23 | 13 | 8 | 44 |
| | DSMART with expert rules | 19 | 11 | 7 | 37 |
| | DSMART with extra soil observations | 22 | 9 | 7 | 38 |
| Independent soil profiles (n=135) | Original DSMART | 11 | 5 | 3.8 | 18.1 |
| | DSMART with expert rules | 10 | 4.4 | 3.7 | 19.8 |
| | DSMART with extra soil observations | 8.2 | 6 | 2.7 | 16.9 |
| Legacy soil profiles (n=755) | Original DSMART | 14 | 7 | 6 | 27 |
| | DSMART with expert rules | 18 | 9 | 7 | 34 |
| | DSMART with extra soil observations | | | | |



| Pixel to pixel validation of STU group | | | | | |
|---|---|---|---|---|---|
| | DSMART approach | Most probable STU | Second most probable STU | Third most probable STU | Total |
| Soil maps (87 150 ha) | Original DSMART | 26 | 13 | 9 | 48 |
| | DSMART with expert rules | 22.5 | 13.7 | 9.7 | 45.9 |
| | DSMART with extra soil observations | 25 | 10 | 7 | 42 |
| Independent soil profiles (n=135) | Original DSMART | 16 | 7 | 4.6 | 27.6 |
| | DSMART with expert rules | 18 | 8.4 | 5.2 | 31.6 |
| | DSMART with extra soil observations | 15 | 8 | 3.8 | 26.8 |
| Legacy soil profiles (n=755) | Original DSMART | 19 | 12 | 9 | 40 |
| | DSMART with expert rules | 23.4 | 15 | 11.8 | 50.2 |
| | DSMART with extra soil observations | | | | |

| Neighbourhood of 3 x 3 validation of STU | | | | | |
|---|---|---|---|---|---|
| | DSMART approach | Most probable STU | Second most probable STU | Third most probable STU | Total |
| Soil maps (87 150 ha) | Original DSMART | 31 | 16 | 14 | 61 |
| | DSMART with expert rules | 29.6 | 19.4 | 13.1 | 62.1 |
| | DSMART with extra soil observations | 28 | 11 | 9 | 48 |
| Independent soil profiles (n=135) | Original DSMART | 15 | 6 | 4.3 | 25.3 |
| | DSMART with expert rules | 17 | 6.7 | 4.8 | 28.5 |
| | DSMART with extra soil observations | 11 | 7 | 3 | 21 |
| Legacy soil profiles (n=755) | Original DSMART | 19 | 10 | 7 | 36 |
| | DSMART with expert rules | 27.9 | 15 | 11.9 | 54.8 |
| | DSMART with extra soil observations | | | | |





*Table 4: Comparison between the size areas covered, number of soil map units, soil type units of the original legacy soil maps and the accuracy achieved in other studies using DSMART algorithm*

| Study | Area (km²) | Map units | Soil type unit | Accuracy |
|---|---|---|---|---|
| Odgers et al (2014) | 68,000 | 1,110 | 72 | 23 |
| Holmes et al. (2015) | 2,500,000 | 5,069 | 73 | 20-22 |
| Chaney et al. (2016) | - | - | - | 17 |
| Møller et al. (2019) | 43,000 | 11-14 | 18-23 | 12-18 |