# Peer review of "Title: Comparing three approaches of spatial disaggregation of legacy soil maps based on DSMART algorithm Authors: Yosra Ellili-Bargaoui1,2, Brendan Philip Malone3, Didier Michot4, Budiman Minasny5, Sébastien Vincent1, Christian Walter<s"

_SOIL, 2019_

## Referee Comment (RC1) · Madlene Nussbaum (Referee) · 2 Jul 2019

The manuscript is relevant as it tackles two very practical problems in completing missing spatial soil information in general: 1) how to fully exploit partly heavily aggregated legacy soil maps and 2) how to include otherwise available knowledge into this process. Two types of knowledge were separately tested (but not combined): soil legacy data and local expert knowledge of the study region. The latter seems a very relevant endeavor as it can reduce reconnaissance survey efforts and drop the costs of creating more accurate maps significantly.

The manuscript is mostly well assembled, logically structured and mostly written in

adequate language.

However, I would like the editor and authors to consider the following remarks:

**1 Novelty**

Three methods are applied to the same study region and their performance to predict soil units (STU) are compared in the manuscript. The first method its the DSMART default algorithm published by Odgers et al. (2014). The second includes actual soil observations. This is new to my knowledge, but also quite straightforward. The innovative part of including expert knowledge stored in DoneSol in a structured way was, however, already published in Vincent et al. (2018, Geoderma). Comparing three methods and evaluating their performance justifies an additional article as long as the approaches are applied in a very sound statistical framework. Here, improvements are recommended (see below).

**2 Introduction**

The introduction should revised. First, it relies on few publications only. Then, it splits the approaches in two groups (L83-34) of which the first group is not advised for the presented study region extent. The actual opposed groups here are not approaches using no covariates (e. g. ordinary kriging – which is an obsolete approach for digital soil mapping as with the spatial coordinates present universal kriging should at least be applied) and approaches using covariates (as e. g. DSMART). For the large study area presented here I would never advise for kriging without covariates. The difference might be made between approaches that use actual observations as response (e. g. DSM in Nussbaum et al. 2018 and many others) while other approaches generate artificial soil

observations from available covariates (this would theoretically not be limited to legacy soil maps).

**3   Covariates not comprehensive**

The authors state that for this landscape waterlogging is very characteristic. However, curvatures or TPI (see detailed comment on L185) representing terrain depressions were only used at one scale/resolution. Was there a reason for that? There are many publications showing the benefit of including a multitude of terrain attributes. Therefore, I suggest to also include other terrain attributes as e.g. MRVBF (multi resolution valley bottom flatness, see Nussbaum et al. 2018 for application and references).

Following this aspect, it is not clear to me that plain DSMART algorithm would actually be outperformed by method 3 which includes expert based rules. There were just not enough covariates included in the model to fully represent the soil forming factors in method 1.

**4   Weighing scheme for approach with legacy soil profiles (method 2)**

The authors should maybe consider to apply a weighting scheme to the response during the model fit for method 2.

The 755 actual observations are mixed with 14 000 artificial observations drawn form the legacy soil map polygons. The artificial observations largely outnumber the "more true" observations. I understand that the random assignment of STU (L212, step 2) in each iteration is only done for the artificial observations while the actual observations stay the same. However, the actual observations most likely "drown" in the abundance

of the artificial ones during model fit. Giving higher weight to the actual observations might increase model performance.

I suggest that the authors at least test a weighing scheme and evaluate its efficiency through e. g. cross-validation (the weighing scheme cannot be selected based on the validation soil data).

**5  Statistical approach**

To train the models the C5.0 decision tree approach was used (CART with some simplification of the rules after tree growth). However, classification and regression trees (CART) are often outperformed by ensemble tree approaches (see e.g. Liess et al. 2012, Liess, M., Glaser, B., and Huwe, B.: Uncertainty in the spatial prediction of soil texture. Comparison of regression tree and Random Forest models, Geoderma, 170, 70–79, doi: 10.1016/j.geoderma.2011.10.010, 2012) more complex methods often yield better results. Usage of ensemble tree methods (e. g. boosted classification trees, cubist with committees or random forest) or other models able to catch complexity (e. g. support vector machines) might improve model performance substantially.

The models trained on artificially generated data are anyway not open to much pedological interpretation. Using a simple single tree approach does not result in any advantage. Ensemble tree methods also allow for covariate importance plots (and partial dependence plots for further interpretation).

**6  Evaluation of model performance**

It remains unclear what is meant by the reported overall accuracy. Most likely the hit rate / percentage correctly allocated STUs was reported. Please specify in the methods

section.

This measure, however, might be hedged (Wilks, 2011, Chapt. 8). Scoring rules should be applied that evaluate the gain of prediction accuracy compared to a random assignment (e.g. pierce skill score, see Wilks, Chapter 8, Statistical methods in the atmospheric sciences, 2011, R Package *verification*). Brier skill score would by suitable for the probabilistic multi-category setting presented here.

With a percentage correct of about 20–30 % it can be expected that a skill score would be as low as 0.1 (interpretation of a skill score: 0: predictions are completely random, 1: perfect predictions, -1: predictions are completely biased to predict the opposite). The properly evaluated model performance is expected to be very low and not much better than a random map generator (to await authors response). Therefore, all three approaches might not justify a map production nor a publication as a success. I am not against failure publications, but they should be discussed as such and possible reasons for the situation and improvements should be given.

**7 Pairwise map comparison, section 2.6**

The authors spent a lot of words/formulas in the manuscript in defining measures to pairwise compare the predicted maps. However, all three maps remain one realization without a claim of being completely valid. The statement of one realization is a bit more similar to the second than the third does not confirm the validity of the predictions. Such a comparison is not meaningful without any further justification/goal. Moreover, one predicted map being more heterogenouous than the other does not mean it is more valid.

I suggest to drop the entire sections or to explicitly justify why comparing the predictions is meaningful.

**8   Unbalanced response**

It seems the response STU categories do not have equal probability distribution. Hence, the nominal response is unbalanced. According to the manuscript (L348) the less frequent STU were rarely or not predicted.

Tree-based methods especially tend to overpredict the majority categories. The prediction is calculated by majority vote in the final tree leaf and minority classes will in most tree leaves be outvoted and not predicted although the tree splits were meaningfully done. The authors should consider to test a sampling scheme that balances the response. Or in case this was used, please specify and put this aspect explicitly in the text.

**9   Detailed comments**

(L: line in the discussion manuscript):

P1L52-53, Abstract: What accuracy measure did you use? Hit rate/percentage correct? Please specify?

L91: Please replace "developed" by "formalized". The approach was already used before (what this publication widely shows).

L119: It is not relevant that the authors used a HPC (it would be, if your article would focus on HPC and DSM). Please consider dropping.

L167-169: As long ans this publication is not accessible: Please consider at least adding the stratification criteria and weights between strata.

L170: Was this "purposive sampling" by expert knowledge of soil surveyors? Please specify.

L173: Incomplete sentence.

L177: A thought on a detail: How exactly did you convert the point data (e. g. point shapefile) to a raster of 50 m resolution? Where there never 2 profiles in the same pixel? Which could be technically possible and asks for resolution of the conflict.

L179, Section 2.3: Original pixel resolution is not given for every dataset. Please consider reporting it here.

L185: Please give a direct citation of the TPI algorithm instead of an application paper. Was it: Jenness, J.: Topographic Position Index (TPI) v. 1.2, http://www.jennessent.com, 2006 ? Moreover, according to Vincent et al. 2018 you did not use the TPI itself, but a TPI based landscape classification (according to Weiss ca. 2001?). A TPI is zero-centered continuous covariate similar to curvature not a categoric covaraiate.

L236: Please try to avoid "extrapolate" without further specification (you mean spatial extrapolation here). Extrapolation outside of the given data value ranges should only be done exceptionally. Better wording would be something like: "From this fitted model we computed predictions for each node of the 50m-grid throughout the study area".

L243: Please explain UTS. (or did you mean STU?)

L243: Please specify what you mean with "This approach..". Method 3 or the work of Vincent et al.?

L254: Please give more details on "a fixed number". How was it determined?

L256: Please specify proportion of what, occurrence count, area?

L256: How many samples from the expert rules and the random set? Please specify.

L258: What do you mean by "a unique". Please consider removing.

L299: What is the difference of regions and zones? Are these e. g. predictions

calculated by method 1 and method 2? Please specify.

L345: For method 2 172 STU were predicted. Is this number correct as the maximum STU is 171?

L380: Please consider replacing "quality" by "uncertainty".

L383-387: Please always report in the same order. Consider using labels as "method 1", "method 2" to ease readability.

L391: Please consider reformulation, e. g. replace "recorded" by "suggested".

All figures: some text is too small.

Figure 1: In the map legend please specify the scale for "Accurate soil maps". Moreover, please change a different color or shape (e. g. triangles) for the red and green dots. Having the same color saturation they are not visible for about 10

Figure 2: Please slightly enlarge the smallest fonts and explain the abbreviations in the figure caption for readers only checking this figure.

Figure 3: Please replace numbers in legend with soil type unit names or at least indicate the general meaning of the numbers in the figure caption.

Figure 4 and 8: One legend is enough (if they contain the same color scheme).

Figure 6: x-axis labels are missing. Please add.

Many thanks to accept my comments, Best regards, M. Nussbaum, BFH-HAFL

---

## Referee Comment (RC2) · Caroline Chartin (Referee) · 24 Sep 2019

General comments

This paper focuses on testing if, and in which way, disaggregating legacy soil map is improved by adding supportive data in the procedure, i.e. soil legacy data and soil-landscape relationships deduced from local expert knowledge. The purpose of this study is important given the lack of accurate soil information in many regions of the world. Those are particularly needed to better face the current process threatening/degrading soils. Moreover, some methods tested here could considerably help diminishing the time and cost for producing new accurate soil maps by reducing field-

work efforts.

In my opinion, the manuscript is mostly well structured, logical, and the language correct.

However, I have some concerns about the approach and the methodology.

Specific comments

My concerns about this study join those highlighted earlier in the discussion by the referee Madlene Nussbaum.

Indeed, the authors proposed to compare three methods of disaggregation, each based on the DSMART algorithm, and to test them on the Ille-et-Vilaine department. As far as I understand, the method 3 was proposed by Vincent et al. in 2018 who applied it to the entire Brittany (which includes the Ille-et-Vilaine department) using the same covariates (at the same resolution) and validation databases used here, but obviously at a bigger extent. Although Vincent et al. (2018) do not detail the results obtained by using the classical version of DSMART (i.e., the Method 1 here), they already visually compared the maps resulting from Methods 1 and 3 on a reduced area of Ille-et-Vilaine.

The maps obtained here and in Vincent et al. (2018) showed that only $\sim$ 20 % of the validation data had been correctly predicted. The authors of the latter study already highlighted that adding the soil-landscape relationships (Method 3) did not substantially improve the results accuracy but tend to produce a more pedologically coherent map. Hence, the authors of Vincent et al. (2018) proposed different coherent ways to optimize the disaggregation procedure and improve its performance (through improving soil data, covariates, and predictive models).

Here, the authors proposed to improve input data by combining DSMART algorithm to legacy soil data. Unfortunately, the legacy soil data are very largely outnumbered by the observations created artificially by the algorithm, limiting greatly their potential

effect on the model performance. In this context, applying Method 2 is almost the same as applying Method 1. A weighing procedure should be implemented in the procedure for the Method 2.

Considering the low performance of the different methods, I suggest the authors to dig in more in improving the procedure as suggested by Vincent et al. (2018) before applying a pairwise map comparative study. For example, the use of the legacy soil data could be optimized in Method 2 as proposed above and more complex and efficient ensemble tree methods could be tested (e.g., random Forest, cforest...) which have many advantages as, among others, integrated validation procedure and clear estimation of the respective variable importance in the model.

Technical corrections

l.143: Please, replace the underscore '_' by the dash '-' in "0_20 m" and "20_50 m".

l. 165-178: §2.2.2. 'Soil validation data'

- As the existing detail maps define one of the three validation datasets, l. 173-174 should be aligned with l.167-172.

- The dataset extracted from 'Sols de Bretagne' is used for validation of M1 and M3 but also used as calibration datasets in M2: it has to be clear somewhere in the text.

- Could you please precise what are the main characteristics considered by an expert to define a STU and how you converted legacy data points and vector maps to raster (l. 176-178)?

l. 277-291: The validation procedure should be more explicit and maybe improved by computing one or two more parameters in order to better apprehend the performance of the models.

l. 179-200: §2.3 'Soil covariates'

Please, could you quickly justify the choice of the covariates used in this procedure,

and maybe make a parallel with the characteristics considered for defining STU?

- The TPI and waterlogging parameters are categorized here. I understand that it facilitates the computation of the soil landscape relationships, but have you try to input the continuous versions of these parameters in the models?

- The landscape unit parameter is an aggregation of vegetation, land use and relief attributes. Why did you prefer to use one aggregated layer instead of more accurate maps about land use, vegetation and relief attributes? Is there a significant correlation between all of these parameters? Is it in order to take into account the landscape morphology at different scales, i.e. main features with the Landscape units and then local features within thanks to more accurate relief attributes layers?

l. 256: Could you precise which proportions of the 18,320 samples used in the Method 3 are derived from expert knowledge and from the random selection implemented in the DSMART algorithm?

Figure 1: Please, reduce the size of the dots or change to triangles. Precise the scale of the detail maps.

Figure 3: Please, could you precise the names of the STU in the legend or in the caption.

Figure 6: Please, add the x-axis labels.

Figure 7: Please, harmonize the covariate names with main text.

I wish my comments would be considered and helpful. I stay available for any discussion.

Best regards, Caroline Chartin

---

## Author Comment (AC1) · 6 Feb 2020

Thank you for taking the time to review our manuscript. We will address the comments and revise the paper accordingly. Below reviewer's comments and our responses.

The manuscript is relevant as it tackles two very practical problems in completing missing spatial soil information in general: 1) how to fully exploit partly heavily aggregated legacy soil maps and 2) how to include otherwise available knowledge into this process. Two types of knowledge were separately tested (but not combined): soil legacy data and local expert knowledge of the study region. The latter seems a very relevant endeavor as it can reduce reconnaissance survey efforts and drop the costs of creating

more accurate maps significantly. The manuscript is mostly well assembled, logically structured and mostly written in adequate language. However, I would like the editor and authors to consider the following remarks:

1 Novelty Three methods are applied to the same study region and their performance to predict soil units (STU) are compared in the manuscript. The first method it is the DSMART default algorithm published by Odgers et al. (2014). The second includes actual soil observations. This is new to my knowledge, but also quite straightforward. The innovative part of including expert knowledge stored in DoneSol in a structured way was, however, already published in Vincent et al. (2018, Geoderma). Comparing three methods and evaluating their performance justifies an additional article as long as the approaches are applied in a very sound statistical framework. Here, improvements are recommended (see below).

2 Introduction The introduction should revised. First, it relies on few publications only. Then, it splits the approaches in two groups (L83-34) of which the first group is not advised for the presented study region extent. The actual opposed groups here are not approaches using no covariates (e. g. ordinary kriging – which is an obsolete approach for digital soil mapping as with the spatial coordinates present universal kriging should at least be applied) and approaches using covariates (as e. g. DSMART). For the large study area presented here I would never advise for kriging without covariates. The difference might be made between approaches that use actual observations as response (e. g. DSM in Nussbaum et al. 2018 and many others) while other approaches generate artificial observations from available covariates (this would theoretically not be limited to legacy soil maps).

RESPONSE: We thank Dr M. Nussbaum for the constructive feedback. As detailed below we have tried to address the reviewer's concerns about the introduction. In the introduction, we tried to present at the beginning the main needs and challenges for improving soil information resolution and scale. These needs deal with solving environmental issues and improving the consideration of soils in management and planning

**SOILD**
strategies at various spatial scales. Moreover, we presented possible approaches that can be used to characterize the spatial distribution of soil information as regard to existing soil data and available environmental covariates. The general approach synthesizes the decision tree for digital soil mapping based on legacy soil data as proposed by Minasny and McBratney 2010 (Figure 1). Hempel et al (2014) also recommend using this workflow to create GlobalSoilMap.net soil property information and generate digital soil maps at high spatial resolution. According to Minasny and McBratney, 2010 "The methods used for digital soil mapping depends on the availability of soil data. The possibilities in the order from the richest to the poorest soil information are: 1. Detailed soil maps with legends and soil point data This is the richest information that can give the best prediction of soil properties. Soil properties can be derived from both soil maps and soil point data. The available methods are: extracting soil properties from soil map using a spatially weighted measure of central tendency, e.g. the mean, spatial disaggregation of soil maps, scorpan kriging and combination of these. An example of such an application is Henderson et al. (2001, 2005) in Australia. 2. Soil point data When soil point data are available, soil properties can be interpolated and extrapolated to the whole area by using a combination of empirical deterministic modelling and a stochastic spatial component. We have called this the scorpan kriging approach. 3. Detailed soil maps with legends When only soil maps are available, we need to extract soil properties from soil maps using some central and distributional concepts of soil mapping units. 4. No data When no data or soil maps exist in area, we will use an approach we call homosoil, which means that we need to estimate the likely soil properties under the observed soil-forming factors or scorpan factors".

On the other hand, in a recent study entitled "Disaggregation of conventional soil maps by generating multi realizations of soil class distribution (case study: Saadat Shahr plain, Iran)" (Jamshidi et al., 2019), the authors emphasize the need of using digital soil mapping approaches, particularly spatial disaggregation of legacy soil data, which considered as the most exhaustive soil information available over large areas. In other related DSMART studies like Odgers et al., 2014, Chaney et al., 2016, the researchers

**SOILD**
have focused on spatial disaggregation approaches of legacy maps and presented the main steps of the DSMART algorithm as well as the structure of the legacy soil data. As suggested, we added more references to illustrate the use of observations and soil points data to calibrate soil prediction model (Malone et al., 2009, Nelson and Odeh, 2009, Abdel-Kader, 2011, Jafari et al., 2013, Kempen et al., 2012, Brungard et al., 2015, Mosleh et al., 2016, Viloria et al., 2016, Nussbaum et al. 2018, Padarian et al., 2019). However, in the literature, only few studies have used legacy soil maps and environmental covariates to generate virtual soil observations to disaggregate legacy maps as done by Odgers et al., 2014, Holmes el 2015, Chaney et al., 2016, Costa et al., 2019, Jamshidi et al., 2019; Moller et al., 2019, Zeraatpisheh et al., 2019.

**3 Covariates not comprehensive**

The authors state that for this landscape waterlogging is very characteristic. However, curvatures or TPI (see detailed comment on L185) representing terrain depressions were only used at one scale/resolution. Was there a reason for that? There are many publications showing the benefit of including a multitude of terrain attributes. Therefore, I suggest to also include other terrain attributes as e.g. MRVBF (multi resolution valley bottom flatness, see Nussbaum et al. 2018 for application and references).

RESPONSE: In our study, the TPI was used at a unique spatial resolution of 50 m for many raisons. Firstly, for running DSMART algorithm, all the environmental covariates must be expressed at the same spatial resolution. In our case, the selected resolution depends mostly on the resolution of the available DEM over the whole area and its accessibility as well. Secondly, in our context the selected resolution allowed to characterize and capture the main variation of topographic and geomorphologic features of our study area. The TPI is based on the upstream drainage network, and therefore it intrinsically integrates the variability of the environment over all of the watersheds and not only on neighboring pixels. Therefore, using multiple resolution of this integrative covariate does not markedly improve the prediction process. As demonstrated in a previous study by Lacoste et al., 2014, using multiple covariates resolution introduce some

SOILD
noise because of the high correlation existing between these variables. This could lead to mis-modelling the drainage network, and consequently the soil deposition areas. In the other hand, the selection of covariates was based on a prior knowledge of the study area and its soil forming factors particularly the parent material and some topographic characteristics like the elevation. The choice of environmental covariates was also based on previous studies carried out over the same study area like Lacoste et al. 2011, Lacoste et al. 2014, Lemercier et al. 2012.

Following this aspect, it is not clear to me that plain DSMART algorithm would actually be outperformed by method 3 which includes expert-based rules. There were just not enough covariates included in the model to fully represent the soil forming factors in method 1.

RESPONSE: When comparing soil map depicting dominant soil type unit (STU) of each soil map unit (SMU) with the three disaggregated soil maps, we observed that disaggregated maps capture the main pattern of soil distribution over the study area. The visual inspection of these maps shows that the original DSMART approach promote the prediction of the dominant soil type unit (STU) with high proportion undependably from soil forming factors. However, local variations and clear internal disaggregation were located in the south part of the study area. The validation results using the three soil data (legacy soil profiles, independent soil profiles and accurate maps) highlight the absence of significant differences between disaggregated maps and almost the same performance of the three DSMART approaches. However, according to a prior pedological expertise and knowledge of the study area, we noticed that soil map derived from DSMART with soil/landscape rules gives more coherent soil type distribution and clear internal disaggregation of SMU with a well-developed hydrographic network using the same soil forming factors. Hence, the contribution of implemented soil /landscape rules were judged according to a prior expert knowledge of the study area and not proven by the validation results. The outperformance of the DSMART with expert based rules approach was not statically confirmed but it is clear that the data min-
ing was able to detect the relationships between soil class and landscape over many realizations.

4 Weighing scheme for approach with legacy soil profiles (method 2)

The authors should maybe consider to apply a weighting scheme to the response during the model fit for method 2. The 755 actual observations are mixed with 14 000 artificial observations drawn from the legacy soil map polygons. The artificial observations largely outnumber the "more true" observations. I understand that the random assignment of STU (L212, step 2) in each iteration is only done for the artificial observations while the actual observations stay the same. However, the actual observations most likely "drown" in the abundance of the artificial ones during model fit. Giving higher weight to the actual observations might increase model performance. I suggest that the authors at least test a weighing scheme and evaluate its efficiency through e. g. cross-validation (the weighing scheme cannot be selected based on the validation soil data).

RESPONSE: It is a good suggestion to give high weight to legacy soil profiles, which represent a small percentage of the virtual observations drown from the legacy soil map polygons. However, the 755 extra soil profiles used to calibrate the model were already used to define the spatial boundaries of legacy polygons. Consequently, giving more weight to soil observations can bias predictions and overestimate the performance of this approach. Maybe the best way would be to use an independent soil dataset with extra soil profiles and giving more weight for the additional soil dataset.

5 Statistical approach To train the models the C5.0 decision tree approach was used (CART with some simplification of the rules after tree growth). However, classification and regression trees (CART) are often outperformed by ensemble tree approaches (see e.g. Liess et al. 2012, Liess, M., Glaser, B., and Huwe, B.: Uncertainty in the spatial prediction of soil texture. Comparison of regression tree and Random Forest models, Geoderma, 170, 70–79, doi: 10.1016/j.geoderma.2011.10.010, 2012) more
complex methods often yield better results. Usage of ensemble tree methods (e.g. boosted classification trees, cubist with committees or random forest) or other models able to catch complexity (e.g. support vector machines) might improve model performance substantially. The models trained on artificially generated data are anyway not open to much pedological interpretation. Using a simple single tree approach does not result in any advantage. Ensemble tree methods also allow for covariate importance plots (and partial dependence plots for further interpretation).

RESPONSE: The objective of this study was not to select the best model that can be implemented in the DSMART algorithm to disaggregate legacy soil polygons as done by Moller et al., 2019 "Improved disaggregation of conventional soil maps". Our study aimed to assess the contribution of soil/landscape rules in the disaggregation procedure of existing legacy soil maps. Most of studies like Odgers et al, 2014, Holmes el 2015, emphasize the need of implementing expert based rules in the original DSMART algorithm in order to improve the performance of prediction of soil types. However, as mentioned by Moller et al., 2019 no study has verified this hypothesis and assessed the real contribution of soil landscape rules in the disaggregation procedure nor how these rules can enhance the spatial characterization of soil distribution. To this end, we applied the same model as Vincent et al., 2018 at large spatial extend and we tried to characterize the differences between disaggregated soil maps generated by each DSMART based approach by using different validation approaches and pairwise comparison method. However, it worthwhile to investigate in futures studies the use of ensemble tree methods in the DSMART algorithm and optimizing the disaggregation process to improve the spatial characterization of soil distribution.

6 Evaluation of model performance It remains unclear what is meant by the reported overall accuracy. Most likely the hit rate / percentage correctly allocated STUs was reported. Please specify in the methods This measure, however, might be hedged (Wilks, 2011, Chapt. 8). Scoring rules should be applied that evaluate the gain of prediction accuracy compared to a random assignment (e.g. pierce skill score, see Wilks, Chap-
ter 8, Statistical methods in the atmospheric sciences, 2011, R Package verification). Brier skill score would by suitable for the probabilistic multi-category setting presented here. With a percentage correct of about 20–30 % it can be expected that a skill score would be as low as 0.1 (interpretation of a skill score: 0: predictions are completely random, 1: perfect predictions, -1: predictions are completely biased to predict the opposite). The properly evaluated model performance is expected to be very low and not much better than a random map generator (to await authors response). Therefore, all three approaches might not justify a map production nor a publication as a success. I am not against failure publications, but they should be discussed as such and possible reasons for the situation and improvements should be given.

RESPONSE: Many studies, which mobilized DSMART algorithm, like Odgers et al., 2014. Holmes et al, Chaney et al 2016, Vincent et al., 2018, Moller et al., 2019, Zeraatpisheh et al., 2019, Jamshidi et al., 2019 have used the term Âń the overall accuracy Âz to report the percentage of soil profiles where observations meet predictions. In this context, the overall accuracy corresponds to the number of correctly predicted classes to the total classes. For example, if we have 755 observations, and we well predict the STU of 200 profiles, the overall accuracy equals to (200/750)\* 100= 26.7%. In our study, the low overall accuracy values are explained by the complexity of the legacy soil data and the high number of STU that contained the soil database. Indeed, the Donesol database contains 171 STU, which are in most of case very similar and differ by some pedological criteria like the clay content or the thickness of some diagnostic horizons. These similarities affect the model performance, particularly where the differences between STU are not easily detected by learning rules. Improving validation results and model performance were discussed in the manuscript, particularly in the sections 4.2 (legacy dataset) and 4.3 (taxonomy similarities). Here, we suggested to simplify the legacy soil data and to create a new soil typology by grouping similar soil types and we also suggest to use taxonomic distance to validate soil maps. In a recent publication entitled "Validation of digital soil maps derived from spatial disaggregation of legacy soil maps" (Ellili-Bargaoui et al, 2019, https://doi.org/10.1016/j.geoderma.2019.113907),

**SOILD**
we developed a validation strategy to validate STU maps, single classification criterion maps (parent material, soil depth, soil natural drainage class, soil type) and continuous soil property maps using an independent validation dataset, selected by stratified random sampling design. Overall, our findings show that we correctly predict single classification criterion with good accuracy measures.

7 Pairwise map comparison,

section 2.6 The authors spent a lot of words/formulas in the manuscript in defining measures to pairwise compare the predicted maps. However, all three maps remain one realization without a claim of being completely valid. The statement of one realization is a bit more similar to the second than the third does not confirm the validity of the predictions. Such a comparison is not meaningful without any further justification/goal. Moreover, one predicted map being more heterogeneous than the other does not mean it is more valid. I suggest to drop the entire sections or to explicitly justify why comparing the predictions is meaningful.

RESPONSE: Disaggregated soil maps were not generated from only one realization but from 100 realizations for both original DSMART and DSMART with extra soil observations approaches and from 50 realizations for DSMART with soil/landscape rules. All realizations were stacked together to compute the probability of occurrence of the 171 STU (Donesol database) at each pixel and then attribute the most probable STU to each elementary pixel. The visual inspection of the three-disaggregated maps shows high similarities and local differences. As validation results do not allowed selecting the best disaggregation approach, we have based on the expert pedological knowledge to choose the best disaggregated map which will be used later to derive soil property maps. These maps are required to calibrate decision support and diagnostic tools needed for sustainable soil-landscape management. Using pairwise comparison of disaggregated maps allowed simultaneously visualizing and locating the main differences between the reference map chosen by the expert (DSMART with expert based rules) and the two other maps. Disaggregated soil maps differ mainly by the numbers SOILD
of regions, which correspond to the spatial delimitation of STU in each complex SMU and the predicted STU at each pixel. Consequently, the pairwise comparison gives a visual support to compare maps and highlights the contribution of expert based rules. For example, we observe that soil landscape rules promote the prediction of hydromorphic soils in the bottom valley area. Almost, similar trend characterizes DSMART with extra soil observations map, particularly in the north part of the study area where extra observations have been collected. Moreover, pairwise comparison method is a new approach, which never has been used before in soil sciences field despite its potentialities. To this end, we decided to keep this section and showing the results of the pairwise comparison of soil maps to illustrate how V-measure method can be used in soil sciences field and help to interpret soil maps differences derived from different methods.

8 Unbalanced response It seems the response STU categories do not have equal probability distribution. Hence, the nominal response is unbalanced. According to the manuscript (L348) the less frequent STU were rarely or not predicted. Tree-based methods especially tend to overpredict the majority categories. The prediction is calculated by majority vote in the final tree leaf and minority classes will in most tree leaves be outvoted and not predicted although the tree splits were meaningfully done. The authors should consider to test a sampling scheme that balances the response. Or in case this was used, please specify and put this aspect explicitly in the text.

RESPONSE: It is a good suggestion to test a sampling scheme that promote the prediction of less frequent STU. In our study, we do not test this approach but we discussed guiding sampling scheme in the section 4.5) (Improvement and future work). It may be a relevant way to improve the disaggregation process and promote the prediction of less frequent and particular STU.

9 Detailed comments (L: line in the discussion manuscript):

P1L52-53, Abstract: What accuracy measure did you use? Hit rate/percentage cor-
rect? Please specify? RESPONSE: The accuracy measure corresponds to the percentage of soil profiles where predictions meet observations. For example, if we have 755 observations, and we well predict the STU of 200 profiles then the overall accuracy equals to (200/750) \* 100= 26.7% As requested, this was clarified and pointed out in the abstract.

L91: Please replace "developed" by "formalized". The approach was already used before (what this publication widely shows).

Revised as suggested developed was replaced by formalized L119: It is not relevant that the authors used a HPC (it would be, if your article would focus on HPC and DSM). Please consider dropping.

Revised as suggested

L167-169: As long as this publication is not accessible: Please consider at least adding the stratification criteria and weights between strata

RESPONSE: This publication is accessible online and entitled "Validation of digital soil maps derived from spatial disaggregation of legacy soil maps" (Ellili-Bargaoui et al, 2019, https://doi.org/10.1016/j.geoderma.2019.113907).

L170: Was this "purposive sampling" by expert knowledge of soil surveyors? Please specify.

RESPONSE: The validation dataset contains 755 legacy soil profiles. These profiles were sampled based on expert knowledge to characterize pedological diversity. This sentence was revised as suggested to point out the purposive sampling strategy followed to collect these profiles.

L173: Incomplete sentence. RESPONSE: Sentence checked and completed.

L177: A thought on a detail: How exactly did you convert the point data (e. g. point shapefile) to a raster of 50 m resolution? Where there never 2 profiles in the same
pixel? Which could be technically possible and asks for resolution of the conflict.

RESPONSE: We used Arc Toolbox from ArcGIS software to create a raster layer from punctual soil observations and we select the assignment type "Most Frequent", and a cell size of 50 m. In our case, we never have 2 profiles in the same pixel.

L179, Section 2.3: Original pixel resolution is not given for every dataset. Please consider reporting it here.

RESPONSE: The original spatial resolution of soil and environmental covariates are as following:

Soil parent material and waterlogging index covariates were predicted in previous studies using machine learning and point dataset at a spatial resolution of 50m. These studies were done before the achievement of the 250,000-soil map of Brittany. For more details, please refer to Lacoste et al (2011) and Lemercier et al (2012).

Gamma-ray spectrometry data was obtained from an airborne geophysical survey in which flying lines were spaced 250–1000 m apart, and measurements were interpolated by kriging to achieve a final data resolution of 250m (Bonijoly et al., 1999).

Land use is a 250 m-pixel size landscape classification resulting from a supervised classification of MODIS (MODerate resolution Imaging Spectroradiometer) imagery (Le Du-Blayo et al., 2008).

The rest of terrain attributes: elevation, slope, Compound Topographic Index (CTI) were directly derived from a DEM at a 50 m-resolution (IGN, 2008).

As requested, we added a supplement information about the original covariate resolution in table 1.

L185: Please give a direct citation of the TPI algorithm instead of an application paper. Was it: Jenness, J.: Topographic Position Index (TPI) v. 1.2, http://www.jennessent.com, 2006 ? Moreover, according to Vincent et al. 2018 you SOILD
did not use the TPI itself, but a TPI based landscape classification (according to Weiss ca. 2001?). A TPI is zero-centered continuous covariate similar to curvature not a categoric covariate.

RESPONSE: As suggested, the reference Jenness, 2006 was added. Like Vincent et al, 2018, we have used a TPI based landscape classification, which classifies the landscape into 5 classes: ridges, upper slopes, steep slopes, gentle slopes, lower slopes and valleys.

L236: Please try to avoid "extrapolate" without further specification (you mean spatial extrapolation here). Extrapolation outside of the given data value ranges should only be done exceptionally. Better wording would be something like: "From this fitted model we computed predictions for each node of the 50m-grid throughout the study area".

RESPONSE: Revised as suggested

L243: Please explain UTS. (or did you mean STU?)

RESPONSE: It was a mistake. It was checked and fixed.

L243: Please specify what you mean with "This approach...". Method 3 or the work of Vincent et al.? + L254: Please give more details on "a fixed number". How was it determined?

RESPONSE: Method 3 is the work of Vincent et al., 2018. For DSMART with expert based rules we used Vincent et al's., 2018 findings, extracted at the Ille-et-Vilaine department. The fixed number drown from each polygon was determined based on the literature (Odgers et al., 2014). For more details, please refer to the article of Vincent et al., 2018.

L256: Please specify proportion of what, occurrence count, area?

RESPONSE: Area proportion. We added area to clarify the random sampling procedure followed. Interactive comment

L256: How many samples from the expert rules and the random set? Please specify.

RESPONSE: The number of samples from the expert rules can be easily deducted. In the line 266 of the manuscript, we specified that for each realization 18,320 samples were generated, where 14,370 virtual points are randomly selected (line 214). Therefore 18,320 - 14,370 = 3950 points were derived from expert knowledge. As requested, this was clarified and pointed out in the manuscript (Line 266-268).

L258: What do you mean by "a unique". Please consider removing.

**RESPONSE:** Revised as suggested

L299: What is the difference of regions and zones? Are these e. g. predictions calculated by method 1 and method 2? Please specify.

RESPONSE: Exactly, it means that prediction calculated by method 1 are called regions and predictions calculated by method 2 are called zones.

L345: For method 2 172 STU were predicted. Is this number correct as the maximum STU is 171?

RESPONSE: For method 2 (DSMART with 755 supplement soil profiles) we predicted 172 STU because to calibrate the model (C5.0) we merge two sources of data: - Virtual soil samples derived from random sampling of legacy polygons to be then assigned to 171 STU (STU contained in the legacy soil data) -755 legacy soil profiles which have been assigned to 172 different STU. Hence, there is an extra STU which not exists in the legacy soil database.

L380: Please consider replacing "quality" by "uncertainty".

Revised as suggested.

L383-387: Please always report in the same order. Consider using labels as "method 1", "method 2" to ease readability.

SOILD
Revised as suggested

L391: Please consider reformulation, e. g. replace "recorded" by "suggested".

Revised as suggested.

All figures: some text is too small. Figure 1: In the map legend please specify the scale for "Accurate soil maps". Moreover, please change a different color or shape (e. g. triangles) for the red and green dots. Having the same color saturation, they are not visible for about 10 Figure 2: Please slightly enlarge the smallest fonts and explain the abbreviations in the figure caption for readers only checking this figure. Figure 3: Please replace numbers in legend with soil type unit names or at least indicate the general meaning of the numbers in the figure caption. Figure 4 and 8: One legend is enough (if they contain the same color scheme). Figure 6: x-axis labels are missing. Please add.

Revised as suggested.

Many thanks for your suggestions that allowed us to improve our paper.
**SOILD**

Figure 1: A decision tree for digital soil mapping based on legacy soil data (Minasny and McBratney, 2010, Hempel et al 2014) Interactive comment

Fig. 1.

---

## Author Comment (AC2) · 6 Feb 2020

Thank you for taking the time to review our manuscript. We will address the comments and revise the paper accordingly. Below reviewer's comments and our responses.

General comments

This paper focuses on testing if, and in which way, disaggregating legacy soil map is improved by adding supportive data in the procedure, i.e. soil legacy data and soil-landscape relationships deduced from local expert knowledge. The purpose of this study is important given the lack of accurate soil information in many regions of

the world. Those are particularly needed to better face the current process threatening/degrading soils. Moreover, some methods tested here could considerably help diminishing the time and cost for producing new accurate soil maps by reducing field work efforts. In my opinion, the manuscript is mostly well structured, logical, and the language correct. However, I have some concerns about the approach and the methodology.

Specific comments My concerns about this study join those highlighted earlier in the discussion by the referee Madlene Nussbaum. Indeed, the authors proposed to compare three methods of disaggregation, each based on the DSMART algorithm, and to test them on the Ille-et-Vilaine department. As far as I understand, the method 3 was proposed by Vincent et al. in 2018 who applied it to the entire Brittany (which includes the Ille-et-Vilaine department) using the same covariates (at the same resolution) and validation databases used here, but obviously at a bigger extent. Although Vincent et al. (2018) do not detail the results obtained by using the classical version of DSMART (i.e., the Method 1 here), they already visually compared the maps resulting from Methods 1 and 3 on a reduced area of Ille-et-Vilaine. The maps obtained here and in Vincent et al. (2018) showed that only âĹij 20 % of the validation data had been correctly predicted. The authors of the latter study already highlighted that adding the soil-landscape relationships (Method 3) did not substantially improve the results accuracy but tend to produce a more pedologically coherent map. Hence, the authors of Vincent et al. (2018) proposed different coherent ways to optimize the disaggregation procedure and improve its performance (through improving soil data, covariates, and predictive models). Here, the authors proposed to improve input data by combining DSMART algorithm to legacy soil data. Unfortunately, the legacy soil data are very largely outnumbered by the observations created artificially by the algorithm, limiting greatly their potential effect on the model performance. In this context, applying Method 2 is almost the same as applying Method 1. A weighing procedure should be implemented in the procedure for the Method 2. Considering the low performance of the different methods, I suggest the authors to dig in more in improving the procedure as suggested

by Vincent et al. (2018) before applying a pairwise map comparative study. For example, the use of the legacy soil data could be optimized in Method 2 as proposed above and more complex and efficient ensemble tree methods could be tested (e.g., random Forest, cforest. . .) which have many advantages as, among others, integrated validation procedure and clear estimation of the respective variable importance in the model.

We thank Dr C. Chartin for the constructive feedback. As detailed below we have tried to address the reviewer's concerns about the methodology followed.

RESPONSE: The suggestion to give high weight to legacy soil profiles, which represent a small percentage of the virtual observations drown from the legacy soil map polygons (Method 2) is very relevant. However, the 755 extra soil profiles used to calibrate the model were already used to define the spatial boundaries of legacy polygons. Consequently, giving more weight to soil observations can bias predictions and overestimate the performance of this approach. May be the best way could be to use an independent soil dataset with extra soil profiles and giving more weight for the additional soil dataset. The objective of this study is not to select the best model that can be implemented in the DSMART algorithm to disaggregate legacy soil polygons as done by Moller et al., 2019 "Improved disaggregation of conventional soil maps". Our study aimed to assess the contribution of soil/landscape rules in the disaggregation procedure of existing legacy soil maps. Most of studies like Odgers et al, 2014, Holmes el 2015, emphasize the need of implementing expert-based rules in the original DSMART algorithm in order to improve the performance of prediction of soil types. However, as mentioned by Moller et al., 2019 no study has verified this hypothesis and assessed the real contribution of soil landscape rules in the disaggregation procedure nor how these rules can enhance the spatial characterization of soil distribution. To this end, we applied the same model as Vincent et al., 2018 at large spatial extend and we tried to characterize the differences between disaggregated soil maps generated by each DSMART based approach by using different validation approaches and pairwise

comparison method.

Technical corrections

l.143: Please, replace the underscore '_' by the dash '-' in "0_20 m" and "20_50 m".

Revised as suggested

l. 165-178: §2.2.2. 'Soil validation data' - As the existing detail maps define one of the three validation datasets, l. 173-174 should be aligned with l.167-172.

Revised as suggested

- The dataset extracted from 'Sols de Bretagne' is used for validation of M1 and M3 but also used as calibration datasets in M2: it has to be clear somewhere in the text.

RESPONSE: As suggested, we added the following sentence to clarify this point in line 297 "In this study, the validation dataset with 755 observations was used to assess the accuracy of digital maps derived from method 1 and method 3 and it was used as additional calibration dataset for method 2". This was also pointed out in the Figure 2, which presents the schematic of DSMART based approaches investigated in our study.

- Could you please precise what are the main characteristics considered by an expert to define a STU and how you converted legacy data points and vector maps to raster (l. 176-178)?

RESPONSE: The STU nomenclature respects the French soil classification system (Baize and Girard, 2008). It reflects different information at the same time like the weathering degree of soil parent material, the redoximorphic conditions, the soil type (referring to the identification of diagnostic horizons depicting pedogenetic processes), and the soil depth. This was clarified in the manuscript (Line 163-167).

We used Arc Toolbox from ArcGIS software to create a raster layer from punctual soil observations using the tool points to raster conversion. In our case, we never have 2 profiles in the same pixel of 50m.

We also used Arc Toolbox from ArcGIS software to create raster maps from accurate maps using the tool polygon to raster conversion and we selected the assignment type "Most Frequent", and a cell size of 50 m.

l. 277-291: The validation procedure should be more explicit and maybe improved by computing one or two more parameters in order to better apprehend the performance of the models.

RESPONSE: Most of studies that applied DSMART algorithm like Odgers et al., 2014, Holmes el 2015, Chaney et al., 2016, Jamshidi et al., 2019; Moller et al., 2019, Zer-aatpisheh et al., 2019 have computed the same validation measures "the overall accu-racy". Moreover, only few studies like Chaney et al., 2016, Vincent et al., 2018 have considered pixel neighbored, as we done in our study, to compute validation measures with some flexibility.

In a recent publication entitled "Validation of digital soil maps derived from spatial disaggregation of legacy soil maps" (Ellili-Bargaoui et al, 2019, https://doi.org/10.1016/j.geoderma.2019.113907), we developed a validation strategy to validate STU maps, single classification criterion maps (parent material, soil depth, soil drainage class, soil type) and continuous soil property maps using an independent dataset, selected by stratified random sampling design.

l. 179-200: §2.3 'Soil covariates' Please, could you quickly justify the choice of the covariates used in this procedure, and maybe make a parallel with the characteristics considered for defining STU?

RESPONSE: The selection of covariates was based on a prior knowledge of the study area and its soil forming factors particularly the parent material and some topographic characteristics like the elevation. This choice was also based on previous studies car-ried out over the same study area like Lacoste et al. 2011, Lacoste et al. 2014, Lemercier et al. 2012. Moreover, some soil covariates particularly soil parent material and soil drainage characteristics are also used to define STU.
- The TPI and waterlogging parameters are categorized here. I understand that it facilitates the computation of the soil landscape relationships, but have you try to input the continuous versions of these parameters in the models?

RESPONSE: Like Vincent et al, 2018, we have used a TPI based landscape classification, which classifies the landscape into 5 classes: ridges, upper slopes, steep slopes, gentle slopes, lower slopes and valleys. In our study, we do not use the continuous version of the TPI to calibrate the model, but we expect found similar results.
- The landscape unit parameter is an aggregation of vegetation, land use and relief attributes. Why did you prefer to use one aggregated layer instead of more accurate maps about land use, vegetation and relief attributes? Is there a significant correlation between all of these parameters? Is it in order to take into account the landscape morphology at different scales, i.e. main features with the Landscape units and then local features within thanks to more accurate relief attributes layers?

RESPONSE: The landscape classification resulting from a supervised classification of MODIS (MODerate resolution Imaging Spectroradiometer) imagery (Le Du-Blayo et al., 2008). This landscape classification particularly focuses on agricultural land use and spatial organization, considering not only land cover and relief, but also elements of the landscape as the network of hedges. It allowed considering the landscape morphology and capturing the main landscape and local feature of our study area. Until now, there is no more accurate exhaustive information on landscape units taking into account spatial organization of the agricultural land.

l. 256: Could you precise which proportions of the 18,320 samples used in the Method 3 are derived from expert knowledge and from the random selection implemented in the DSMART algorithm?

RESPONSE: The number of samples from the expert rules can be easily deducted. In the line 266 of the manuscript, we specified that for each realization 18, 320 samples were generated, where 14,370 virtual points are randomly selected (line 220). Therefore 18,320 – 14,370 = 3950 points were derived from expert knowledge. As requested, this was clarified and pointed out in the manuscript (Line 266-268).

Figure 1: Please, reduce the size of the dots or change to triangles. Precise the scale of the detail maps.

Revised as suggested.

Figure 3: Please, could you precise the names of the STU in the legend or in the caption.

Revised as suggested.

Figure 6: Please, add the x-axis labels.

The x-axis labels correspond to STU ordered by increasing cumulative area as mentioned on the Figure.

Figure 7: Please, harmonize the covariate names with main text.

Revised as suggested.

Many thanks for your suggestions that allowed us to improve our paper.